

# Sequential flows by irrelevant operators

### Christian Ferko[1] and Savdeep Sethi[2]

**1** Center for Quantum Mathematics and Physics (QMAP),
Department of Physics & Astronomy, University of California,
Davis, CA 95616, USA

**2** Enrico Fermi Institute & Kadanoff Center for Theoretical Physics,
University of Chicago, Chicago, IL 60637, USA

## Abstract

We explore whether one can $T\overline{T}$ deform a collection of theories that are already $T\overline{T}$-deformed. This allows us to define classes of irrelevant deformations that know about subsystems. In some basic cases, we explore the spectrum that results from this procedure and we provide numerical evidence in favor of modular invariance. We also study the flow of the classical Lagrangian for free bosons and free fermions under successive deformations. Some of the models found by sequentially flowing are likely to have interesting holographic interpretations.

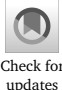

# 1 Introduction

The $T\overline{T}$ deformation is an interesting irrelevant deformation of quantum field theories in two dimensions [1–3]. The $T\overline{T}$ operator is constructed from the following quadratic combination of stress-energy tensors,

$$T\overline{T}(x) = \lim_{y \to x} \left( T^{\mu\nu}(x)T_{\mu\nu}(y) - T^{\mu}_{\ \mu}(x)T^{\nu}_{\ \nu}(y) \right). \tag{1}$$

It is universal in the sense that it requires little more than translation invariance. It is natural to wonder what other tractable irrelevant deformations might exist. Analogues of the form $J_1 \overline{J}_2$ have been studied where $J_1$ and $J_2$ are conserved currents, including higher spin currents. This work is more exploratory in nature: our aim is to see what happens when we deform theories with subsystems that are already themselves deformed. We will provide evidence for the existence of theories which do not follow from the original reasoning that leads to the $T\overline{T}$ deformation. For example, the leading irrelevant deformation is not a scalar operator built from conserved currents of the theory.

The nature of a $T\overline{T}$-deformed theory is currently mysterious. Quantizing the theory on a cylinder of radius $R$ gives an energy spectrum which satisfies the inviscid Burgers' equation:

$$\frac{\partial}{\partial \lambda} E_n(R, \lambda) = E_n(R, \lambda)\frac{\partial}{\partial R} E_n(R, \lambda) + \frac{P_n(R)^2}{R}. \tag{2}$$

Here $E_n$ are the energies and $P_n$ are the quantized momenta. If $P_n = 0$, the equation reduces to $\partial_\lambda E_n = E_n \partial_R E_n$; in the absence of shocks, this equation admits an implicit solution:

$$E_n(R, \lambda) = E_n\left(R + \lambda E_n(R, \lambda), 0\right). \tag{3}$$

On the other hand if the seed theory is conformal, equation (2) can be solved explicitly for general $P_n$,

$$E_n(\lambda) = \frac{R}{2\lambda}\left( \sqrt{1 + \frac{4\lambda E_n}{R} + \frac{4\lambda^2 P_n^2}{R^2}} - 1 \right). \tag{4}$$

For the good sign of the deformation ($\lambda > 0$), the high-energy density of states is Hagedorn and the energies are real. This signals some kind of non-locality in the theory, perhaps analogous to the non-locality found in string theory. Note that the ground state energy, $E_0$, for a unitary CFT is negative. For sufficiently large $\lambda$, the ground state energy will become complex so there is a bound:

$$\lambda \le \frac{R}{4|E_0|}. \tag{5}$$

Beyond this inequality, the high-energy density of states has passed the point of the Hagedorn phase transition and the torus partition function is typically no longer convergent.

For the bad sign ($\lambda < 0$), the situation is considerably more mysterious. Integrating the inviscid Burgers' equation to find the deformed spectrum always encounters a shock singularity for an infinite number of sufficiently large initial energies. This happens regardless of how small one chooses $\lambda$. After encountering the singularity, the energy given by the formal solution (4) becomes complex and multi-valued. It is not at all clear that using the implicit solution (3) is sensible after reaching the singularity. At this point one needs some prescription to define the spectrum, assuming the theory exists at all. Often in the fluids literature, a physically motivated conservation equation weaker than the inviscid Burgers' equation (2) is

imposed [4]. It would be very interesting if some analogue of that procedure can be found for quantum field theory.

In this discussion, we will not need to assume the theory makes sense for $\lambda < 0$, but we will occasionally use deformations with this sign in intermediate steps, or even in a final flow, as long as two criteria are satisfied. The first criterion is some reasonable prescription for determining the final energy spectrum. The second criterion is that the final deformed energies are real for some range of deformation parameters.

One of the basic features of local quantum field theory is that given a collection of theories, one can tensor the theories together. Imagine tensoring two local quantum field theories together. One might wonder whether we can define an irrelevant deformation that couples the two theories together in a way that knows about the subsystems. Something like $T_1 \overline{T}_2$ rather than the original $T\overline{T}$ deformation, which is agnostic to any subsystem structure.

This turns out to be closely related to the following question: can one $T\overline{T}$ deform a collection of theories with each already $T\overline{T}$-deformed? In one case, the answer is clear. For a single theory, we can continuously perturb by the good sign $T\overline{T}$ operator because that is how the deformation is essentially defined. Since the deformation preserves translation invariance, there is no issue with defining the operator at each point along the flow. As a first case, we explore sequential deformations of a single theory in section 2.

To define $T_1 \overline{T}_2$, let us restrict to seed theories which are conformal so we can use the explicit energy formula (4) for the seed energy spectrum. Take $\text{CFT}^1_{\lambda_1}$ and $\text{CFT}^2_{\lambda_2}$, where each theory is deformed with parameter $\lambda_1$ or $\lambda_2$, respectively. Tensor these two theories together. We should be able to now deform the tensor product $\text{CFT}^1_{\lambda_1} \otimes \text{CFT}^2_{\lambda_2}$ to obtain a theory which we denote as $\left\{ \text{CFT}^1_{\lambda_1} \otimes \text{CFT}^2_{\lambda_2} \right\}_{\lambda_3}$. The first order in $(\lambda_1, \lambda_2, \lambda_3)$ deforming operator is,

$$\lambda_3 \left[ (T_1 + T_2)(\overline{T}_1 + \overline{T}_2) \right] + \lambda_1 T_1 \overline{T}_1 + \lambda_2 T_2 \overline{T}_2 \,. \tag{6}$$

In writing this operator, we are only using the undeformed initial stress-energy tensors. If we choose $\lambda_3 = -\lambda_1 = -\lambda_2$ then this operator is

$$\lambda_3 \left[ T_1 \overline{T}_2 + \overline{T}_1 T_2 \right] \,. \tag{7}$$

It is not at all clear that the operator in (7) exists beyond first order in $\lambda_3$. The individual operators $T_1$ and $T_2$ do not have any immediate definition once one $T\overline{T}$ deforms the combined system because only the energy and momentum of the total system is conserved. Yet the procedure of sequentially deforming that we described would seem to define some theory, whose leading order deformation might be taken to be (7) perhaps only in the special limit where $\lambda_3 = -\lambda_1 = -\lambda_2$ are infinitesimal. Visually, the procedure we have in mind is depicted in Figure 1.

At this stage, we can try to determine the energy spectrum of $\left\{ \text{CFT}^1_{\lambda_1} \otimes \text{CFT}^2_{\lambda_2} \right\}_{\lambda_3}$. Even though $\text{CFT}^1_{\lambda_1}$ might have a complex energy spectrum for $\lambda_1 < 0$, the additional deformation of the combined theory might restore real energies for some range of the deformation parameter. That is what seems to happen when studying combinations like $J\overline{T} + T\overline{T}$, where $J\overline{T}$ alone always has complex energies for any choice of deformation parameter [5–8]. Let us very briefly summarize what we find for the case of a bipartite system:

- For $\lambda_1 > 0, \lambda_2 > 0, \lambda_3 > 0$, we find a real energy spectrum with a bound on how large the flow parameters can become before the ground state energy goes complex. This is completely analogous to the constraint on the good sign deformation of a single theory given in (5). We also present some numerical evidence in favor of modular invariance of the resulting spectrum. This case is explored in section 3.1.

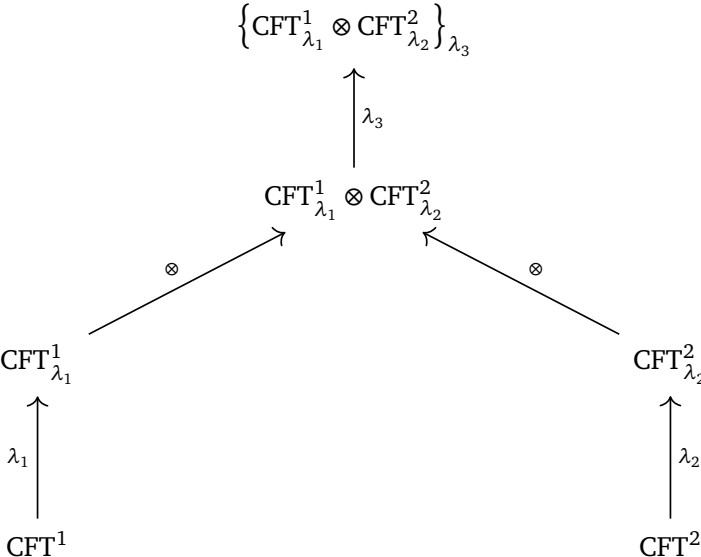

Figure 1: Sequentially deforming two CFTs.

- For $\lambda_1 > 0, \lambda_2 > 0, \lambda_3 < 0$, we always find complex energies. Depending on the relative amounts of good sign versus bad sign flows, there can be a finite or infinite number of complex energies. This case is explored in section 3.2.

- For $\lambda_1 < 0, \lambda_2 < 0, \lambda_3 > 0$, we find that in specific cases like $\lambda_3 = r|\lambda_1|$ with $\lambda_1 = \lambda_2$ the spectrum can be all real when $r \geq 2$. There is a more intricate and interesting phase structure for $r < 2$, described in section 3.3, when the theory has complex energies.

We also explore cases with $\lambda_2 = 0$ in section 3.4. In section 4, we study the flow equation for the classical Lagrangian in the case of two free bosons and in the case of two free fermions. While a single flow takes a free boson to the gauge-fixed Nambu-Goto action, the second flow generates a kind of interacting theory of multiple strings. It would be interesting to explore the relation of this deformation with other $T\overline{T}$-inspired deformations of string theory, like the case studied in [9].

**Models of potential holographic interest**

While our discussion here is mainly focused on quantum field theory, we cannot resist commenting on some specific cases that are of potential interest for holography. Take a specific example of $AdS_3/CFT_2$ duality. A possible holographic interpretation of $CFT_2$ deformed by the wrong sign $T\overline{T}$ flow has been offered in [10]. The interpretation is a kind of cutoff AdS spacetime. However, this deformed $CFT_2$ has an infinite number of complex energies, which makes its interpretation as a field theory unclear. If one tensors together two such theories and then deforms the combination with a sufficiently large good sign flow then our analysis suggests the resulting theory is free of any immediate pathologies. It might be possible to interpret this procedure in terms of wormhole physics along the lines of [11–13]. For such an endeavor, it is likely one will need a more complete understanding of the holographic interpretation of the good sign $T\overline{T}$-deformed $CFT_2$.[1]

---

[1]Some progress has been made in interpreting the holographic good sign $T\overline{T}$ deformation as a change of boundary conditions for the $3d$ bulk fields, either in metric [14] or Chern-Simons variables [15]. See also [16–19] for related analyses in the dimensionally reduced setting where boundary conditions are modified for $2d$ bulk fields dual to a $(0+1)$-dimensional $T\overline{T}$-deformed quantum mechanics.

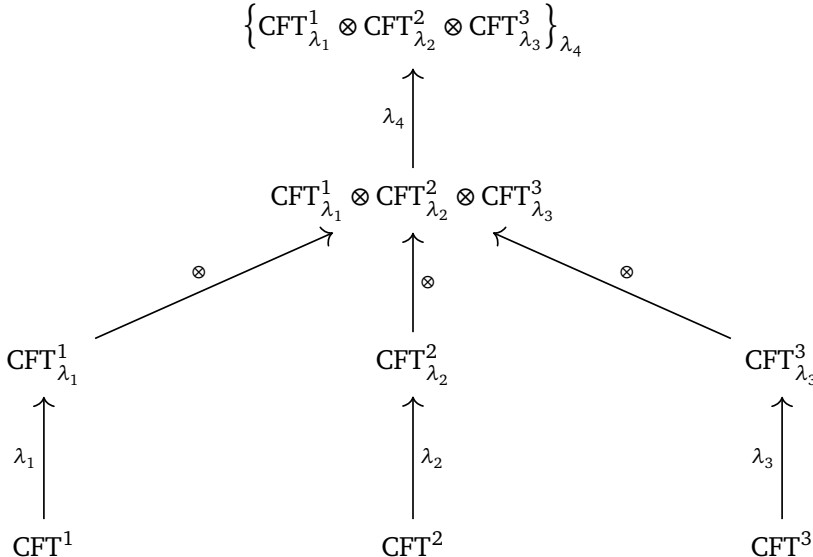

Figure 2: Deforming the tensor product of three 'level 1' deformed theories.

A more robust holographic proposal has been offered in [20, 21]. This involves a kind of single trace analogue of the good sign $T\overline{T}$ deformation, although a precise definition of the deforming operator is unknown. The holographic interpretation involves changing the spacetime from asymptotically AdS to asymptotically linear dilaton. One could again consider the wrong sign for the single trace deformation. In the bulk, this has been discussed in [22]. The field theory should have the same pathologies as the conventional wrong sign double-trace $T\overline{T}$ deformation. One could try a similar cure for this theory, as described above, by tensoring two such theories together and flowing the combination by a sufficient amount of good sign double trace $T\overline{T}$. We should stress that this case is interesting in its own right simply from a field theory perspective since it involves a mix of single trace and double trace deformations.

**Conventions**

As a matter of convention, we will denote dimensionless energies and parameters by variables with a tilde. Explicitly for a theory on a cylinder of size $R$,

$$\widetilde{\lambda} = \frac{\lambda}{R^2}, \qquad \widetilde{E} = ER, \qquad \widetilde{\mathcal{E}} = \mathcal{E}R. \tag{8}$$

**Future directions**

It seems likely that we are only scratching the surface of a large class of non-local theories. For example, one could relax constraints like Lorentz invariance, or consider higher spin deformations. Even if one restricts to Lorentz invariant theories and only considers sequential flows by $T\overline{T}$ operators, there are many interesting possibilities.

Imagine, for example, that we begin with three seed theories. The direct analogue of the bipartite case is to flow each one individually and then flow the tensor product. This is pictured in figure 2. We can view the final step as deforming the tensor product of three 'level 1' deformed theories.

In this case, we could alternatively perform the procedure depicted in figure 3. The final step can be viewed as deforming the tensor product of a 'level 2' deformed theory with a 'level 1' deformed theory.

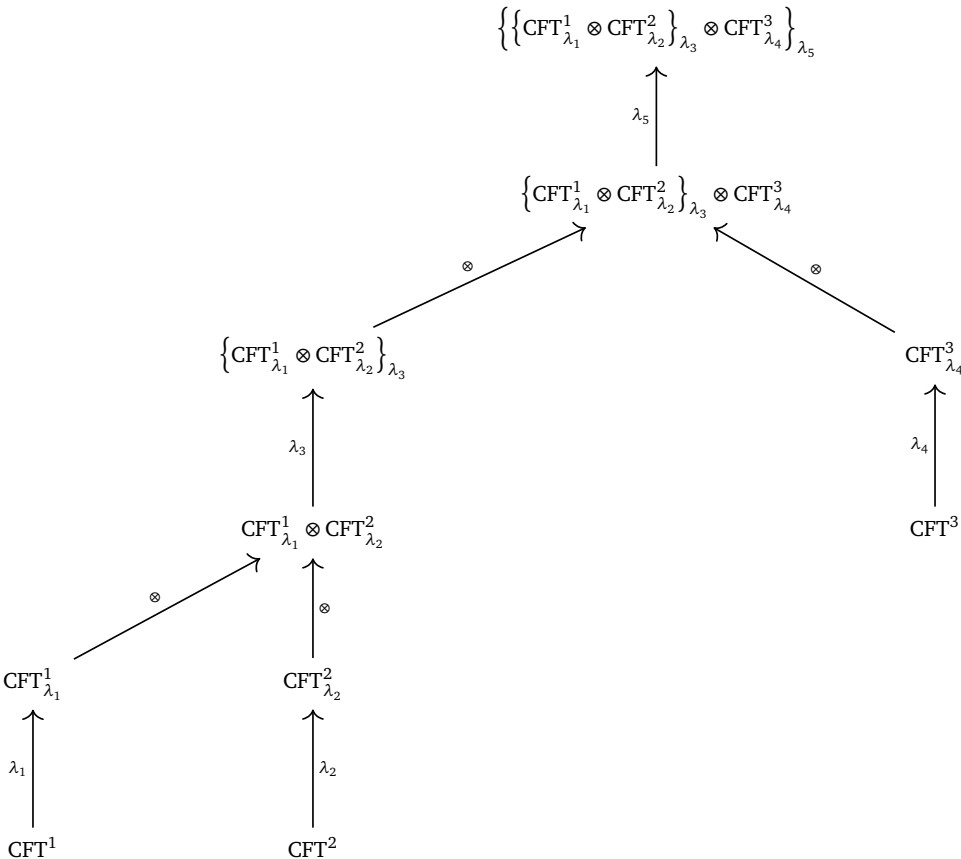

Figure 3: Deforming the tensor product of a 'level 2' deformed theory with a 'level 1' deformed theory.

This kind of construction can clearly be extended to $N$ theories in many ways with potentially interesting large $N$ limits. The most straightforward generalization is to flow the tensor product of $N$ 'level 1' deformed theories along the lines of figure 4.

## 2 Deforming a Single Theory

We want to understand what kind of energy spectrum results from solving (2) for examples like the sequence of $T\bar{T}$ deformations depicted in figure 1. As a warm up case, let us first consider a single theory deformed by two successive $T\bar{T}$ deformations. Although this is a well-studied example, the structure seen in this case will help illuminate what we find in examples that involve multiple systems. We will examine three cases from most conservative and most likely to result in a unitary theory to more speculative.

For simplicity, let us consider the zero momentum sector using a seed theory which is a CFT. To avoid confusion, we will introduce three different symbols for the energies at each step of the deformation process:

$$e_n \xrightarrow{\lambda_1} E_n \xrightarrow{\lambda_2} \mathcal{E}_n. \tag{9}$$

That is, $e_n = e_n(R)$ are the energies in the totally undeformed CFT, $E_n = E_n(R, \lambda_1)$ are the energies after the first deformation step, and $\mathcal{E}_n = \mathcal{E}_n(R, \lambda_1, \lambda_2)$ are the final energies after both deformations.

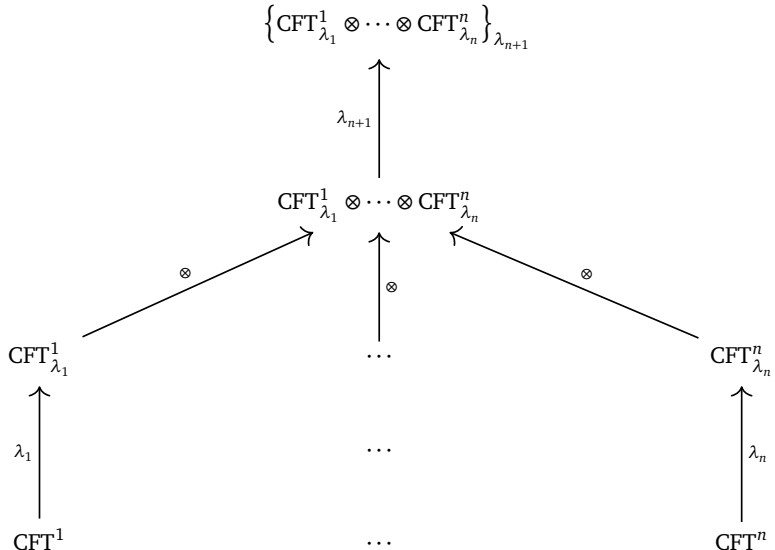

Figure 4: Deforming the tensor product of $N$ 'level 1' deformed CFTs.

## 2.1 Sequential good sign deformations

As we saw in equation (4), after the first deformation step by parameter $\lambda_1 > 0$, the energies are given by

$$E_n(\lambda_1) = \frac{R}{2\lambda_1} \left( \sqrt{1 + \frac{4\lambda_1 e_n}{R}} - 1 \right). \tag{10}$$

We now deform the theory with energies (10) again, this time by parameter $\lambda_2 > 0$. Since the new initial theory is no longer conformal, we cannot simply use the result (4) again to find the final energies after the second deformation step. However, since we are restricting to the zero momentum sector, we are free to use the implicit solution (3) to the inviscid Burgers' equation, which we reproduce here for convenience:

$$\mathcal{E}_n(R, \lambda_1, \lambda_2) = E_n\big(R + \lambda_2 \mathcal{E}_n, \lambda_1\big). \tag{11}$$

Because (11) instructs us to replace all instances of the cylinder radius $R$, we must restore the dependence of the CFT energies $e_n$ on $R$. In any unitary CFT one has states with energies,

$$e_n = \frac{\Delta_n + \overline{\Delta}_n - \frac{c}{12}}{R} \equiv \frac{\alpha_n}{R}, \tag{12}$$

where $(\Delta_n, \overline{\Delta}_n)$ are the conformal dimensions of local operators, and we have introduced the notation $\alpha_n$ for brevity. We are also assuming $c = \bar{c}$ for simplicity. Then the intermediate energies with all $R$-dependence made explicit are given by

$$E_n(\lambda_1) = \frac{R}{2\lambda_1} \left( \sqrt{1 + \frac{4\lambda_1 \alpha_n}{R^2}} - 1 \right). \tag{13}$$

Equations (11) and (13) give rise to the implicit relation

$$\mathcal{E}_n = \frac{R + \lambda_2 \mathcal{E}_n}{2\lambda_1} \left( \sqrt{1 + \frac{4\lambda_1 \alpha_n}{(R + \lambda_2 \mathcal{E}_n)^2}} - 1 \right), \tag{14}$$

which can be rearranged as

$$(2\lambda_1 + \lambda_2)\mathcal{E}_n + R = \sqrt{(R + \lambda_2\mathcal{E}_n)^2 + 4\lambda_1\alpha_n}. \tag{15}$$

Squaring both sides of this constraint then gives a quadratic equation for the final energies $\mathcal{E}_n$ whose solution is

$$\mathcal{E}_n = \frac{R}{2(\lambda_1 + \lambda_2)}\left(\sqrt{1 + \frac{4(\lambda_1 + \lambda_2)\alpha_n}{R^2}} - 1\right). \tag{16}$$

We see that (16) is exactly of the form (13) except with the deformation parameter $\lambda_1$ replaced by the sum $\lambda_1 + \lambda_2$. In particular, first deforming by $\lambda_1 > 0$ and then deforming by $\lambda_2 > 0$ is the same as deforming by the sum $\lambda_1 + \lambda_2$ all at once.

## 2.2 Good sign followed by bad sign

As before, flowing first with the good sign gives energies $E_n$ with the square root form (10). We now flow by $\lambda_2 < 0$. Do we get a real sensible spectrum? In the analogy with fluid mechanics, the flow in $\lambda$ is a flow in time. Viewed this way, the question is how far back can we flow in 'time' before we hit a singularity. Certainly if $\lambda_2$ is much larger than $\lambda_1$ we expect the theory to behave like bad sign $T\bar{T}$. The issue is whether any amount of backward flow is problematic or a finite amount is permissible.

The implicit solution (3) of the Burgers' equation is the undeformed seed energy evaluated at a radius that is energy-dependent: $\tilde{R} = R + \lambda_2\mathcal{E}_n(R, \lambda_1, \lambda_2)$. Following the discussion in [3, 23], one kind of singularity develops when $\partial_{\tilde{R}}R = 0$. This occurs when

$$1 - \lambda_2\frac{\partial E_n}{\partial x}(x, \lambda_1)\bigg|_{x=\tilde{R}_c} = 0, \tag{17}$$

where the critical radius is located at $R_c = \tilde{R}_c - \lambda_2 E(\tilde{R}_c, \lambda_1)$. Solving for this critical radius in the specific case of a two step flow with $\lambda_1 > 0$ and $\lambda_2 < 0$ gives,

$$\tilde{R}_c^2 = -\frac{\alpha_n(2\lambda_1 + \lambda_2)^2}{\lambda_1 + \lambda_2}, \qquad 2\lambda_1 + \lambda_2 < 0, \tag{18}$$

$$R_c^2 = -4\alpha_n(\lambda_1 + \lambda_2). \tag{19}$$

Note that there is no solution for $\tilde{R}_c$ unless $2\lambda_1 + \lambda_2 < 0$. We hit a shock singularity when $R_c$ has a positive real solution so we want to restrict to $R > R_c$.

This is one condition for a good implicit solution. If one starts with CFT data and flows once, this is sufficient because the only singularity of the initial CFT energy in the complex $R$-plane is a pole at $R = 0$. Since we are looking at multi-step flows, our initial data has a more complicated analytic structure. For example, after the $\lambda_1$ flow the initial data has square root branch points seen in (13). There is also no remaining pole singularity at $R = 0$. In general this is a difficult problem to study analytically [24]. Our primary tool for exploring solutions of the inviscid Burgers' equation will be numerics.

**Avoiding any shock region**

From (16) with positive energy $\alpha_n > 0$, we find the bound

$$|\lambda_2| \le \lambda_1. \tag{20}$$

At the point of equality, we have flowed forward and backward by the same amount. It seems reasonable that we have arrived back at the undeformed spectrum in that case. Note there is no bound from (18) because we never reach a sufficiently large $\lambda_2$. The other bound follows from considering the ground state $\alpha_0 < 0$. For the initial flow by $\lambda_1$, we had a bound that $\lambda_1 \leq \frac{R^2}{4|\alpha_0|}$. The most conservative approach is that we impose this strong constraint and completely avoid any shock region. In this case, the final constraints are $|\lambda_2| \leq \lambda_1 \leq \frac{R^2}{4|\alpha_0|}$.

**Entering and exiting a shock region**

There is another interesting possibility in this two step flow. Suppose we permit ourselves to travel past the singularity in the initial flow forward by taking $\lambda_1 > \frac{R^2}{4|\alpha_0|}$. We still assume that (10) applies giving a complex multi-valued deformed ground state energy at the first step. We could simply declare that our prescription for treating the flow back by $\lambda_2$ corresponds to using the implicit solution again, as we have done in the conservative analysis. In this case the backward flow might 'cure' the complex ground state energy.

We can check whether this is sensible by taking either complex root for the energy of the ground state in the shock region as initial data for the flow backward:

$$E_0(\lambda_1) = \frac{R}{2\lambda_1}\left(\pm i \left|\sqrt{1 - \frac{4\lambda_1|\alpha_0|}{R^2}}\right| - 1\right), \qquad \lambda_1 > \frac{R^2}{4|\alpha_0|}. \tag{21}$$

In the region where the solution (13) gives real energies, there is no ambiguity in the branch of the square root. Demanding that $\lambda_1 \to 0$ give the initial energy fixes the branch to be the positive root. Once we cross the branch point and the energy becomes complex, we have to impose a prescription about how to continue past the singularity into the shock region.

There are two choices of root given in (21). The implicit equation for the flow back by an amount $|\lambda_2|$ depends on the choice of root. To recover a real energy, one must flow back out of the shock region, which requires:

$$|\widetilde{\lambda}_2| \geq \widetilde{\lambda}_1 - \frac{1}{4|\alpha_0|}. \tag{22}$$

With this prescription, we preserve the full spectrum of energies satisfying (16) with no constraint on $\lambda_1$. The only price we pay is that small energies might be complex until $\lambda_2 < 0$ satisfies (22). We are still left with the question of defining the theory with a finite number of complex energies if we do not satisfy (22) but do satisfy $\lambda_1 + \lambda_2 \geq 0$. If we fail to satisfy even $\lambda_1 + \lambda_2 \geq 0$ then we are back in the bad sign situation of an infinite number of complex energies. Perhaps additional ingredients along the lines of [25] might result in a well-defined theory for cases with complex energies. The most conservative option is to simply avoid the shock region entirely.

## 2.3 Bad sign followed by good sign

The final case to consider is to first flow by $\lambda_1 < 0$ followed by a flow with $\lambda_2 > 0$. Let us take the same approach as our prior discussion, and try to use the implicit equation to define these sequential flows. Regardless of the magnitude of $\lambda_1$, the high energy states are largely complex after the first flow. Let us take

$$\lambda_2 = r|\lambda_1|, \tag{23}$$

and ask what happens for different ranges of $r$.

**The $r < 1$ phase**

From the implicit solution (16), we expect most high energy states to remain complex. We simply have not flowed 'forward' enough by positive $\lambda_2$ to cure the complex spectrum. Numerics confirm this picture.

**The $r \geq 1$ phase**

For $r = 1$, we expect the backward then forward flow to return us to the initial undeformed theory. The solution to the two step flow (16) shows this is the case as long as we are careful about correlating the implicit equation with the branch of the square root determining the complex energy. In this case, the only bound is not to flow too far forward and make the ground state complex,

$$(r-1)|\lambda_1| \leq \frac{R^2}{4|\alpha_0|}. \tag{24}$$

Otherwise we have to again deal with a theory with a finite number of complex energies.

Although this was a straightforward algebraic exercise, it demonstrates that a $T\overline{T}$ deformation by positive $\lambda$ can cure a spectrum with an infinite number of complex energies, at least in this simple case. One might have thought that a theory with infinitely many complex energies is an unsuitable seed, and that deforming it with any kind of operator would generically lead to another sick theory. However, we have now checked that applying a $T\overline{T}$ deformation to this pathological seed theory can actually reverse the pathology and generate a final deformed theory with a reasonable spectrum; in this case as long as $\lambda_2 > |\lambda_1|$.

Finally we note that the analysis of this section assumed that $P_n = 0$ so that we could use the implicit solution to the Burgers' equation. However, the full solution with non-zero $P_n$ in (4) has an additional term proportional to $\lambda^2 P_n^2$ in the argument of the square root. Since this extra term is strictly non-negative, it can only improve the behavior of the deformed spectrum, in the sense that states which have real energies for $P_n = 0$ will also have real energies when $P_n \neq 0$.

## 3 Deforming Multiple Theories

Next we will repeat the simplified analysis of section 2 in the case where we tensor together $T\overline{T}$-deformed systems as a first step and then deform by the total $T\overline{T}$ operator of the combined system as the second step. This is how we can generate a deformation like $T_1\overline{T}_2$ that knows about subsystems.

As in the preceding discussion, we will restrict to the zero momentum sector for simplicity and consider a seed theory which is the tensor product of two CFTs:

$$\text{CFT}_{\text{seed}} = \text{CFT}_1 \otimes \text{CFT}_2. \tag{25}$$

The undeformed energies of $\text{CFT}_{\text{seed}}$ will be written as $e_{n,m}$. Each such energy is the sum of two energy eigenvalues, one in $\text{CFT}_1$ and one in $\text{CFT}_2$:

$$e_{n,m} = e_n^{(1)} + e_m^{(2)}. \tag{26}$$

The energies $e_n^{(1)}$, $e_m^{(2)}$ take the form (12), so we will introduce constants $\alpha_n, \beta_m$ and write

$$e_n^{(1)} = \frac{\alpha_n}{R}, \qquad e_m^{(2)} = \frac{\beta_m}{R}. \tag{27}$$

Now we apply a $T_1\overline{T}_1$ deformation with parameter $\lambda_1$, only to the theory $\mathrm{CFT}_1$ with energies $e_n^{(1)}$. Likewise, we apply a $T_2\overline{T}_2$ deformation with parameter $\lambda_2$ to theory $\mathrm{CFT}_2$. The total deformed theory is still a tensor product of the two deformed CFTs, and thus its energy levels are given by the sum of the deformed energies in the two tensor product factors. We write these total deformed energies as

$$E_{n,m}(R,\lambda_1,\lambda_2) = \frac{R}{2\lambda_1}\left(\sqrt{1+\frac{4\lambda_1\alpha_n}{R^2}}-1\right) + \frac{R}{2\lambda_2}\left(\sqrt{1+\frac{4\lambda_2\beta_m}{R^2}}-1\right). \tag{28}$$

For the last deformation step, we will take the tensor product theory with energies (28) as our seed and perform a total $T\overline{T}$ deformation by parameter $\lambda_3$, with $T$ constructed from the overall stress-energy tensor of the combined system. Denote the energies of this final deformed theory by $\mathcal{E}_{n,m}$. Because we are restricting to the zero momentum sector, these energies satisfy the implicit relation

$$\mathcal{E}_{n,m}(R,\lambda_1,\lambda_2,\lambda_3) = E_{n,m}\Big(R+\lambda_3\mathcal{E}_{n,m},\lambda_1,\lambda_2\Big). \tag{29}$$

Using the expression (28) for $E_{n,m}$, this gives the constraint

$$\mathcal{E}_{n,m} = \frac{R+\lambda_3\mathcal{E}_{n,m}}{2\lambda_1}\left(\sqrt{1+\frac{4\lambda_1\alpha_n}{\big(R+\lambda_3\mathcal{E}_{n,m}\big)^2}}-1\right) + \frac{R+\lambda_3\mathcal{E}_{n,m}}{2\lambda_2}\left(\sqrt{1+\frac{4\lambda_2\beta_m}{\big(R+\lambda_3\mathcal{E}_{n,m}\big)^2}}-1\right). \tag{30}$$

For choices of parameters such that a solution exists, equation (30) can be solved for $\mathcal{E}_{n,m}$ by a computer algebra system, although the general result is quite unwieldy and not especially illuminating. It is more tractable if we consider some special cases. We will try to order these cases again roughly from more conservative to less conservative.

## 3.1 All good sign deformations

The most conservative situation would be all good sign flows: $\lambda_1,\lambda_2,\lambda_3 > 0$. To avoid entering the shock region on the first flow, we restrict $\lambda_1$ and $\lambda_2$ as in (5) so that the deformed ground state energies, $\alpha_0$ and $\beta_0$ respectively, remain real.

**High-energy behavior**

Let us first examine the high-energy behavior in this case when both $\alpha_n$ and $\beta_m$ are very large. In this limit,

$$E_{n,m} \sim \sqrt{\frac{\alpha_n}{\lambda_1}} + \sqrt{\frac{\beta_m}{\lambda_2}} - \frac{R}{2\lambda_1} - \frac{R}{2\lambda_2} + \frac{R^2}{8\sqrt{\alpha_n}\lambda_1^{3/2}} + \frac{R^2}{8\sqrt{\beta_m}\lambda_2^{3/2}} + \dots \tag{31}$$

Superficially, we might expect that only the leading two terms in (31) are needed to determine the high-energy behavior of $\mathcal{E}_{n,m}$. However, this is not the case. When solving the implicit equation (29) for $\mathcal{E}_{n,m}$, we replace $R$ with $R+\lambda_3\mathcal{E}_{n,m}$ which means the remaining terms in (31) contribute at the same order as the leading two terms. We can still use the implicit solution (30) to determine $\mathcal{E}_{n,m}$ in a power series in $\lambda_3$ around $\lambda_3 = 0$:

$$\mathcal{E}_{n,m} = \frac{R}{2}\frac{\lambda_1+\lambda_2-\lambda_2\sqrt{1+\frac{4\alpha_n\lambda_1}{R^2}}-\lambda_1\sqrt{1+\frac{4\beta_m\lambda_2}{R^2}}}{-\lambda_1\lambda_2+\frac{\lambda_2\lambda_3}{2}\left(-1+\frac{1}{\sqrt{1+\frac{4\alpha_n\lambda_1}{R^2}}}\right)+\frac{\lambda_1\lambda_3}{2}\left(-1+\frac{1}{\sqrt{1+\frac{4\beta_m\lambda_2}{R^2}}}\right)} + \dots$$

$$\to \frac{2\lambda_2\sqrt{\alpha_n\lambda_1}+2\lambda_1\sqrt{\beta_m\lambda_2}-R(\lambda_1+\lambda_2)}{2\lambda_1\lambda_2+\lambda_1\lambda_3+\lambda_2\lambda_3} + \dots \tag{32}$$

The arrow denotes the result when we take the high-energy limit for $\alpha_n$ and $\beta_m$. This is again a Hagedorn spectrum at high energies characterized by the square root dependence on $\alpha_n$ and $\beta_m$.

**High-energies for** $CFT_2$

Now we can turn to the case where the seed energy $\beta_m$ is taken very large with $\alpha_n$ fixed but otherwise unconstrained. In this case, the general expression for the energy is complicated and not particularly illuminating so let us further simplify by taking,

$$\lambda_1 = \lambda_2 = \lambda_3 = \lambda. \tag{33}$$

The seed energies then take the form,

$$E_{n,m} \sim \frac{R}{2\lambda}\left(\sqrt{1 + \frac{4\lambda\alpha_n}{R^2}} - 1\right) + \sqrt{\frac{\beta_m}{\lambda}} - \frac{R}{2\lambda}. \tag{34}$$

This is very similar to the two step flow we studied in section 2.1. Solving the implicit equation gives deformed energies of the form,

$$\mathcal{E}_{n,m} \sim \frac{2\left(R + \sqrt{\beta_m\lambda}\right)}{15\lambda}\sqrt{1 + \frac{15\lambda\alpha_n}{\left(R + \sqrt{\beta_m\lambda}\right)^2}} + \frac{8}{15}\sqrt{\frac{\beta_m}{\lambda}} - \frac{7R}{15\lambda}. \tag{35}$$

This deformed energy has a similar form to the seed energy (34) with a change in the effective radius $R \to R + \sqrt{\beta_m\lambda}$.

At the expense of a more complicated formula, we can relax (33) and consider

$$\lambda_1 = \lambda_2 = \lambda, \tag{36}$$

with arbitrary $\lambda_3 > 0$. In this case, we find

$$\mathcal{E}_{n,m} \sim \frac{1}{4\lambda^2 + 8\lambda\lambda_3 + 3\lambda_3^2}\Big(4\sqrt{\beta_m\lambda^3} + 4\sqrt{\beta_m\lambda}\lambda_3 - 4\lambda R - 3\lambda_3 R$$
$$-2\sqrt{\lambda\left\{\beta_m\lambda_3^2 + \alpha_n\left(4\lambda^2 + 8\lambda\lambda_3 + 3\lambda_3^2\right) + 2\sqrt{\beta_m\lambda}\lambda_3 R + \lambda R^2\right\}}\Big). \tag{37}$$

The expression (37) reduces to (35) when $\lambda_3 = \lambda$ as should be the case. By examining the square root of (37) we can extract an interesting feature: for large $\beta_m$ we can flow forward by $\lambda_3$ as far as we like even if $\alpha_n = \alpha_0$ is the ground state. Said differently: tensoring the deformed ground state of $CFT_1$ with a deformed high-energy state of $CFT_2$ can cure the tachyon, or complex energy, we might have expected from just flowing $CFT_1$ forward alone.

**The diagonal spectrum**

There is one additional case that admits a nice analytic solution. Take $\lambda_1 = \lambda_2 = \lambda$. There could be seed energies where $\alpha_n = \beta_m = \alpha$; for example, if $CFT_1 = CFT_2$ then all $\alpha_n = \beta_n$. For this diagonal component of the spectrum, the input data for the $\widetilde{\lambda}_3$ flow is simple and we can write out an analytic solution for the two-step deformed energies,

$$\mathcal{E}_{n,m} = \frac{R}{(\lambda + 2\lambda_3)}\left(\sqrt{1 + \frac{4\alpha(\lambda + 2\lambda_3)}{R^2}} - 1\right). \tag{38}$$

These energies become complex when $\lambda + 2\lambda_3$ exceeds $\left|\frac{R^2}{4\alpha}\right|$ when $\alpha$ is negative. This is not surprising because we have simply multiplied the deformed negative energy of $CFT_1$ by a factor of 2 and continued flowing. If we had taken $N$ copies of $\{CFT_1\}_\lambda$ and considered negative energy $\alpha$ in each copy, there would be a bound on $\lambda_3$ of the form $\lambda + N\lambda_3 \leq \frac{R^2}{4|\alpha|}$ to avoid a complex energy. As one final check, note that for large $\alpha$ we recover the expression (32) with $\lambda_1 = \lambda_2 = \lambda$ as long as we expand (38) to leading order in $\lambda_3$.

**The ground state**

The next case of qualitative interest is the ground state given by (28) with both $\alpha_0$ and $\beta_0$ negative. How far forward can we flow by $\lambda_3$ before the ground state energy now goes complex? At least intuitively we still expect it to become complex for some sufficiently large $\lambda_3$. This is clear from the formula for the diagonal spectrum (38) applied to a negative energy state.

Explicit formulae, however, become quite complicated when either $\alpha_0 \neq \beta_0$ or $\lambda_1 \neq \lambda_2$. It seems more useful to examine a few cases numerically to see how things change qualitatively. To present the numerical results, it is convenient to make the energies and parameters dimensionless using $R$ with the convention given in (8) that dimensionless quantities are denoted with a tilde.

As a first case, however, we can at least demonstrate that complex energies develop at *some* sufficiently large value of $\widetilde{\lambda}_3$ using an asymptotic analysis. To do this, we assume that $\widetilde{\lambda}_3 \gg \widetilde{\lambda}_1, \widetilde{\lambda}_2$ and expand the constraint equation (30), keeping only the leading contribution at large $\widetilde{\lambda}_3$. The result is

$$\widetilde{\mathcal{E}}_{n,m}^2 = \frac{\alpha_n + \beta_m}{\widetilde{\lambda}_3}. \tag{39}$$

Up to terms which are subleading at large positive $\widetilde{\lambda}_3$, we see that $\widetilde{\mathcal{E}}_{n,m}^2$ has the same sign as $\alpha_n + \beta_m$. In particular the deformed ground state energy is purely imaginary at this order. Although this asymptotic analysis does not tell us the value of $\widetilde{\lambda}_3$ at which complex energies first appear, it does demonstrate that we cannot maintain a real ground state energy at arbitrarily large values of $\widetilde{\lambda}_3$.

Some numerical results are presented in table 1. When the initial ground state energies $4(\alpha_0, \beta_0) = (-1, -1)$ then the maximum values of $(\widetilde{\lambda}_1, \widetilde{\lambda}_2)$ are $(1, 1)$ before one of the initial seed energies goes complex. The maximum value of $\widetilde{\lambda}_3$ is approximate aside from two exceptional cases where an analytic result is possible. Note that when either $\widetilde{\lambda}_1$ or $\widetilde{\lambda}_2$ approach their critical values, the amount $\widetilde{\lambda}_3$ that we can further flow forward appears to go to zero. Finally we list the resulting ground state energy for the maximum $\widetilde{\lambda}_3$.

As a final sanity check, we can take a look at a range of energies for $\beta_m$ with $\alpha_n = \alpha_0$. We expect no strange behavior for positive $\beta_m$ and some numerical checks appear to confirm that belief.

**Evidence for modular invariance**

To close this discussion of good sign flows, we will provide some numerical evidence in favor of modular invariance of the resulting energy spectrum. For this numerical investigation we consider two copies of the $c = 1$ free boson CFT. If the free boson is compact with radius $\hat{r}$, the CFT energies and momenta are given by

$$E_{CFT} = \frac{m^2}{4\hat{r}^2} + n^2\hat{r}^2 + \hat{N} + \hat{M} - \frac{1}{12}, \qquad P = mn + \left(\hat{N} - \hat{M}\right), \tag{40}$$

Table 1: A table listing the approximate maximum $\widetilde{\lambda}_3$ for several cases along with the resulting ground energy.

| $4(\alpha_0, \beta_0)$ | $(\widetilde{\lambda}_1, \widetilde{\lambda}_2)$ | Max $\widetilde{\lambda}_3$ | $4\times$Energy |
|---|---|---|---|
| $(-1, -1)$ | $(0.5, 0.5)$ | $\frac{1}{4} = 0.25$ | $-4$ |
| $(-1, -1)$ | $(0.5, 0.9)$ | $0.06$ | $-3.24$ |
| $(-1, -1)$ | $(0.5, 0.99)$ | $0.00629$ | $-3.18$ |
| $(-1, -1)$ | $(0.5, 0)$ | $0.302$ | $-3.41$ |
| $(-1, -2)$ | $(0.5, 0.25)$ | $\frac{1}{6} \sim 0.167$ | $-6$ |
| $(-1, -2)$ | $(0.5, 0.45)$ | $0.038$ | $-5.26$ |
| $(-1, -2)$ | $(0.5, 0.49)$ | $0.0077$ | $-5.18$ |
| $(-1, -2)$ | $(0.5, 0)$ | $0.21$ | $-4.54$ |

where $m$ is the momentum quantum number, $n$ is the winding and $(\hat{N}, \hat{M})$ are the oscillator excitations. To simplify calculations, we chose the self-dual radius $\hat{r} = \frac{1}{\sqrt{2}}$ for both $\text{CFT}_1$ and $\text{CFT}_2$ so that the ground state is the only state with negative energy in each CFT. In computing the partition function,

$$Z(\tau) = \sum_{m,n,\hat{N},\hat{M}} e^{2\pi i \tau_1 P} e^{-2\pi \tau_2 E} \,, \tag{41}$$

with $\tau_2 > 1$, the result will be dominated by the ground state.

The most interesting check is the modular S-transformation which sends

$$\tau_2 \longrightarrow \frac{1}{\tau_2} \,. \tag{42}$$

The modular transformation properties of each deformation parameter $\widetilde{\lambda}$ are determined by the radius of the cylinder used to make $\lambda$ dimensionless; see, [26, 27], for example. This means,

$$\widetilde{\lambda} \longrightarrow \frac{\widetilde{\lambda}}{|c\tau + d|^2} \,, \qquad \begin{pmatrix} a & b \\ c & d \end{pmatrix} \in SL(2, \mathbb{Z}), \tag{43}$$

and in our case $(\widetilde{\lambda}_1, \widetilde{\lambda}_2, \widetilde{\lambda}_3)$ are each transformed according to (43). In table 2, we have listed the numerical values of the partition function with $\tau_1 = 0$ for various cutoffs on the sums appearing in (41) along with choices $\tau_2$ and the following choice of deformation parameters

$$\left(\widetilde{\lambda}_1, \widetilde{\lambda}_2, \widetilde{\lambda}_3\right) = (0.1, \, 0.1, \, 0.5). \tag{44}$$

For example, cutoff=2 includes $50,625$ energies which each require a separate numerical solution of the inviscid Burgers' equation. We cannot just numerically solve the implicit equation because that solution only applies to zero momentum states.

If the theory is modular invariant, the value of the partition function should agree with the value at $\frac{1}{\tau_2}$ as long as we transform the three deformation parameters in accord with (43). Even with the relatively low quantum numbers for momentum, winding and oscillators that we included, there is quite good agreement between the partition function and its S-dual value. It would be very interesting to see whether the analytic proof of modular invariance developed in [26, 27] for deforming CFTs can be extended to these more general theories.

Table 2: The value of the partition function for different values of $\tau_2$. The winding and momentum $(m, n)$ run from -cutoff to cutoff, while the oscillator numbers $(\hat{N}, \hat{M})$ run from 0 to cutoff.

| $\tau_2$ | cutoff | $Z$ | $\tau_2$ | cutoff | $Z$ |
|---|---|---|---|---|---|
| 1.2 | 1 | 5.20 | $\frac{1}{1.2}$ | 1 | 5.14 |
| 1.2 | 2 | 5.20 | $\frac{1}{1.2}$ | 2 | 5.19 |
| 1.5 | 1 | 6.37 | $\frac{1}{1.5}$ | 1 | 6.16 |
| 1.5 | 2 | 6.37 | $\frac{1}{1.5}$ | 2 | 6.31 |
| 1.75 | 1 | 8.08 | $\frac{1}{1.75}$ | 1 | 7.60 |
| 1.75 | 2 | 8.08 | $\frac{1}{1.75}$ | 2 | 7.91 |
| 2 | 1 | 10.55 | $\frac{1}{2}$ | 1 | 9.49 |
| 2 | 2 | 10.55 | $\frac{1}{2}$ | 2 | 10.14 |

## 3.2   Sequential flows with $\widetilde{\lambda}_1, \widetilde{\lambda}_2 > 0$ and $\widetilde{\lambda}_3 < 0$

We now turn to another case described in the introduction that motivated this analysis. We want to flow forward by $\widetilde{\lambda}_1 > 0$ and $\widetilde{\lambda}_2 > 0$ separately and then flow the combined resulting theory backward by $\widetilde{\lambda}_3$. The general case is complicated; here we want to establish existence of a reasonable spectrum in any single example so let us restrict to,

$$\widetilde{\lambda}_1 = \widetilde{\lambda}_2 = \widetilde{\lambda} > 0, \qquad \widetilde{\lambda}_3 = -r\widetilde{\lambda}, \qquad r \geq 0. \tag{45}$$

The possible values for the parameter $r$, if any, compatible with a real spectrum will determine what kind of operators like $T_1 \overline{T}_2$, along the lines of (6), we can define this way. Intuition from flowing a single theory forward then backward, described in section 2.2, would suggest that we *can* get a reasonable spectrum.

**The diagonal spectrum**

As a further simplification, let us assume that $\text{CFT}_1 = \text{CFT}_2$. We then have the usual bound on $\widetilde{\lambda} \leq \frac{1}{4|\alpha_0|}$ if we wish to keep the ground state energy real after the first flow. We want to examine how large $r$ can become while still keeping the energies real. For the diagonal spectrum $\alpha_n = \beta_n$ we can use the solution we found earlier, which we reproduce here in terms of dimensionless parameters:

$$\widetilde{\mathcal{E}}_{\text{diag}} = \frac{-1 + \sqrt{1 + 4\alpha\widetilde{\lambda}(1 - 2r)}}{(1 - 2r)\widetilde{\lambda}}. \tag{46}$$

From this we see that $r > \frac{1}{2}$ looks like a bad sign flow with large $\alpha > 0$ becoming complex. For $r < \frac{1}{2}$ there is no obvious pathology and this diagonal spectrum appears to be well-behaved. In this case, the leading irrelevant operator is given by,

$$\lambda \left\{ (1 - r)\left(T_1 \overline{T}_1 + T_2 \overline{T}_2\right) - r\left(T_1 \overline{T}_2 + \overline{T}_1 T_2\right) \right\}, \qquad r < \frac{1}{2}. \tag{47}$$

There is an interesting question of what is happening for $r = \frac{1}{2}$. In this case, the solution to the implicit equation gives $\widetilde{\mathcal{E}}_{\text{diag}} = 2\alpha$, so long as $\widetilde{\lambda} < \frac{1}{\alpha}$, which is confirmed by a numerical

investigation. That is, the deformed diagonal spectrum returns exactly to the undeformed diagonal spectrum for $r = \frac{1}{2}$ and sufficiently small $\widetilde{\lambda}$. We can extend this discussion of the diagonal spectrum to $N$ copies of $\text{CFT}_1$. The bound changes to $r \leq \frac{1}{N}$.

**The off-diagonal spectrum**

Now we would like to explore some features of the off-diagonal spectrum, $\alpha_n \neq \beta_m$, for the case of 2 copies of $\text{CFT}_1$. Because we are flowing backward by by $r\widetilde{\lambda}$, we might have thought any sickness should be visible in the high energy spectrum. This intuition turns out to be wrong. When both $\alpha_n$ and $\beta_m$ are very large, we can use formula (32), which is accurate for small $\widetilde{\lambda}_3$ and thus small $r$, to find

$$\widetilde{\mathcal{E}}_{n,m} \sim \frac{\sqrt{\alpha_n \widetilde{\lambda}} + \sqrt{\beta_m \widetilde{\lambda}} - R}{\widetilde{\lambda}(1-r)} \,. \tag{48}$$

This shows that the high-energy spectrum is free of pathologies at least for small $r$ where the expression (48) is valid.

The other case that needs investigating is when $\alpha_n$ and $\beta_m$ differ substantially. Specifically we can take the ground state $\alpha_0 < 0$ and some $\beta_m > 0$. Here we find a surprise which we did not see for the case of a single theory. Let us revisit the implicit equation we are trying to study:

$$\widetilde{\mathcal{E}}_{0,m} = \frac{1 - r\widetilde{\lambda}\widetilde{\mathcal{E}}_{0,m}}{2\widetilde{\lambda}} \left( \sqrt{1 - \frac{4\widetilde{\lambda}|\alpha_0|}{\left(1 - r\widetilde{\lambda}\widetilde{\mathcal{E}}_{0,m}\right)^2}} - 1 \right) + \frac{1 - r\widetilde{\lambda}\widetilde{\mathcal{E}}_{0,m}}{2\widetilde{\lambda}} \left( \sqrt{1 + \frac{4\widetilde{\lambda}\beta_m}{\left(1 - r\widetilde{\lambda}\widetilde{\mathcal{E}}_{0,m}\right)^2}} - 1 \right). \tag{49}$$

What happens in this case, which did not happen in the case of a single theory, is that the first square root of (49) can become imaginary for large $\widetilde{\mathcal{E}}_{0,m}$ regardless of how small one chooses $r$. Numerics seems to confirm that there are a finite number of complex energies for generic $r$.

To see this graphically, we have plotted the real spectrum for the case of $\widetilde{\lambda} = \frac{1}{2}$ in Figures 5 and 6. In both cases, we begin with ground state energies $\alpha_0 = \beta_0 = -\frac{1}{4}$ and then assume an evenly-spaced discrete spectrum where the gaps between adjacent energy levels is $\frac{1}{10}$ so that $\alpha_{n+1} - \alpha_n = 0.1$. In both cases, almost all of the energies are real except for a small strip that has been excised when one of the energies is negative and the other is moderate and positive. These excluded strips are visible as ragged edges that have been cut off on either boundary of the plots. For this combination of flows, there is a slim chance that exceptional solutions exist where the seed CFT has a special spectrum and one tunes $\widetilde{\lambda}$ and $r$ to specific values. Such a theory exist, should it exist, would be isolated.

In hindsight, the existence of complex energies seems reasonable. For $r = \frac{1}{2}$, the diagonal spectrum returns to its undeformed value. This includes the ground state. However, the off-diagonal spectrum is definitely changed. It is hard to see how such a spectrum could remain compatible with modular invariance.

## 3.3 Sequential flows with $\widetilde{\lambda}_1, \widetilde{\lambda}_2 < 0$ and $\widetilde{\lambda}_3 > 0$

Now we reverse the order of the flows and first flow backward into the shock region and then flow forward. We will use the same simplifying assumption of $\text{CFT}_1 = \text{CFT}_2$ and $\widetilde{\lambda}_1 = \widetilde{\lambda}_2 = \widetilde{\lambda} < 0$ with $\widetilde{\lambda}_3 = r|\widetilde{\lambda}|$. Is there any range of $r$ for which the resulting spectrum is real?

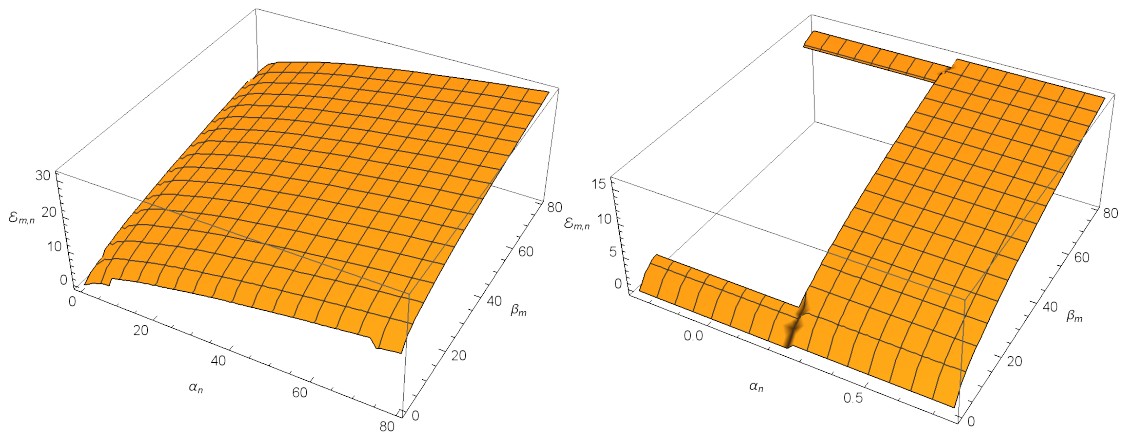

Figure 5: A plot of the deformed energies $\mathcal{E}_{n,m}$ as a function of the undeformed energies $(\alpha_n, \beta_m)$, where $\widetilde{\lambda}_1 = \widetilde{\lambda}_2 = \frac{1}{2}$ and $r = \frac{1}{4}$ so that $\widetilde{\lambda}_3 = -\frac{1}{8}$. The left plot shows the deformed spectrum for $\alpha_n, \beta_m$ ranging from $-\frac{1}{4}$ to 80. The right plot zooms onto the region $-\frac{1}{4} \le \alpha_n \le 1$, where there is a window of complex energies that are not plotted.

**The diagonal spectrum**

There is no immediate natural bound on $\widetilde{\lambda}$ since any amount of backward flow generates complex energies. Let us first examine the diagonal spectrum with $\alpha_n = \beta_n$:

$$\widetilde{\mathcal{E}}_{\text{diag}} = \frac{-1 + \sqrt{1 + 4\alpha|\widetilde{\lambda}|(2r-1)}}{(2r-1)|\widetilde{\lambda}|}. \tag{50}$$

As in section 3.2, there is an exceptional case $r = \frac{1}{2}$ where the diagonal spectrum appears to return to its undeformed value, $\widetilde{\mathcal{E}}_{\text{diag}} = 2\alpha$, so long as $|\widetilde{\lambda}| < \frac{1}{\alpha}$. To prevent high energy states from becoming complex we require $r \ge \frac{1}{2}$. To keep the ground state energy real we also require $2r \le \frac{1}{4|\alpha_0\widetilde{\lambda}|} + 1$. This is a fairly weak bound since $|\widetilde{\lambda}|$ can be very small. As in the previous example, the diagonal spectrum looks quite reasonable. For $N$ copies of $\text{CFT}_1$ rather than 2 copies we replace

$$(2r-1) \rightarrow (Nr-1). \tag{51}$$

**The off-diagonal spectrum**

Now let us examine what is happening for the off-diagonal spectrum. The implicit equation takes the form,

$$\widetilde{\mathcal{E}}_{n,m} = \frac{1 + r|\widetilde{\lambda}|\widetilde{\mathcal{E}}_{n,m}}{2|\widetilde{\lambda}|}\left(1 - \sqrt{1 - \frac{4|\widetilde{\lambda}|\alpha_n}{\left(1 + r|\widetilde{\lambda}|\widetilde{\mathcal{E}}_{n,m}\right)^2}}\right) + \frac{1 + r|\widetilde{\lambda}|\widetilde{\mathcal{E}}_{n,m}}{2|\widetilde{\lambda}|}\left(1 - \sqrt{1 - \frac{4|\widetilde{\lambda}|\beta_m}{\left(1 + r|\widetilde{\lambda}|\widetilde{\mathcal{E}}_{n,m}\right)^2}}\right). \tag{52}$$

The square roots can become imaginary only if $\alpha_n$ or $\beta_m$ is positive. Assume both $\alpha_n > 0$ and $\beta_m > 0$ which should be close to a worst case. If we set $\alpha_n = 0$ and solve, we find the analytic result corresponding to flowing a single system

$$\widetilde{\mathcal{E}}(\alpha_n = 0) = \frac{1}{2|\widetilde{\lambda}|(r-1)}\left(1 - \sqrt{1 + 4\beta_m(r-1)}\right). \tag{53}$$

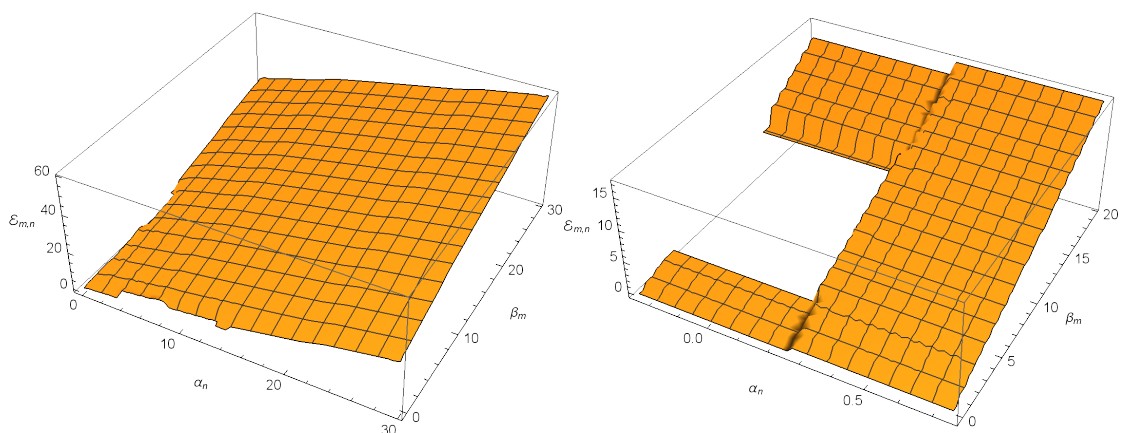

Figure 6: A plot of the deformed energies $\mathcal{E}_{n,m}$ with $\widetilde{\lambda}_1 = \widetilde{\lambda}_2 = \frac{1}{2}$ and $r = \frac{1}{2}$. Again the left plot shows a wide view of the spectrum and the right plot zooms into the region where some energies are excised because the solution to the implicit equation is complex.

This requires $r > 1$ strengthening the constraint seen from the diagonal spectrum.

The last bound we find is something we see numerically; namely, that $r \geq 2$. Unlike the case of good sign followed by bad sign, for this range of $r$ there does appear to be a completely real spectrum. In table 3, we have listed some numerical results for the energies in the zero momentum sector with various choices of $\widetilde{\lambda}$ and $r$.

Table 3: A table showing the numerical solutions for the deformed energies $\widetilde{\mathcal{E}}_{n,m}$ for various choices of dimensionless undeformed energies $(\alpha_n, \beta_m)$, $|\widetilde{\lambda}|$, and $r$. In all cases we take $R = 1$.

| $(\alpha_n, \beta_m)$ | $|\widetilde{\lambda}|$ | $r$ | Energy | $(\alpha_n, \beta_m)$ | $|\widetilde{\lambda}|$ | $r$ | Energy |
|---|---|---|---|---|---|---|---|
| $(-\frac{1}{4}, -\frac{1}{4})$ | $\frac{1}{8}$ | 2 | $-0.558$ | $(-\frac{1}{4}, -\frac{1}{4})$ | $\frac{1}{8}$ | 2.25 | $-0.571$ |
| $(-\frac{1}{4}, 1)$ | $\frac{1}{8}$ | 2 | $0.732$ | $(-\frac{1}{4}, 1)$ | $\frac{1}{8}$ | 2.25 | $0.717$ |
| $(-\frac{1}{4}, 100)$ | $\frac{1}{8}$ | 2 | $24.561$ | $(-\frac{1}{4}, 100)$ | $\frac{1}{8}$ | 2.25 | $22.293$ |
| $(-\frac{1}{4}, -\frac{1}{4})$ | $\frac{1}{4}$ | 2 | $-\frac{2}{3}$ | $(-\frac{1}{4}, -\frac{1}{4})$ | $\frac{1}{4}$ | 2.25 | $-0.739$ |
| $(-\frac{1}{4}, 1)$ | $\frac{1}{4}$ | 2 | $0.705$ | $(-\frac{1}{4}, 1)$ | $\frac{1}{4}$ | 2.25 | $0.680$ |
| $(-\frac{1}{4}, 100)$ | $\frac{1}{4}$ | 2 | $18.097$ | $(-\frac{1}{4}, 100)$ | $\frac{1}{4}$ | 2.25 | $16.356$ |

One way to argue for the bound $r \geq 2$ is as follows: suppose that we consider very high energy states in the final deformed theory so that $\widetilde{\mathcal{E}}_{n,m} \gg 1$. This corresponds to states in the undeformed theory with either $\alpha_n \gg 1$, or $\beta_m \gg 1$, or both. In order for the arguments of the square roots in (52) to remain positive, the ratios

$$\frac{4\alpha_n}{r^2 |\widetilde{\lambda}|^2 \widetilde{\mathcal{E}}_{n,m}^2}, \qquad \frac{4\beta_m}{r^2 |\widetilde{\lambda}|^2 \widetilde{\mathcal{E}}_{n,m}^2}, \tag{54}$$

must remain smaller than 1. For simplicity, we will take the simultaneous limits $\alpha_n \to \infty$, $\beta_m \to \infty$, $\widetilde{\mathcal{E}}_{n,m} \to \infty$ with the ratios $\frac{\alpha_n}{\widetilde{\mathcal{E}}_{n,m}^2}$ and $\frac{\beta_m}{\widetilde{\mathcal{E}}_{n,m}^2}$ held fixed and finite. To leading order, the

implicit equation (52) in this limit can be written as

$$\widetilde{\mathcal{E}}_{n,m}(1-r) = -\frac{1}{2\widetilde{\lambda}}\left(\sqrt{\widetilde{\mathcal{E}}_{n,m}^2 r^2 \widetilde{\lambda}^2 - 4\alpha_n \widetilde{\lambda}} + \sqrt{\widetilde{\mathcal{E}}_{n,m}^2 r^2 \widetilde{\lambda}^2 - 4\beta_m \widetilde{\lambda}}\right). \tag{55}$$

This can be converted into a quartic equation for $\widetilde{\mathcal{E}}_{n,m}$ which has four roots:

$$\widetilde{\mathcal{E}}_{n,m} = \pm\sqrt{\frac{(r-1)(\alpha_n+\beta_m) \pm \sqrt{4\alpha_n\beta_m - 8r\alpha_n\beta_m + r^2(\alpha_n+\beta_m)^2}}{(1-3r+2r^2)\widetilde{\lambda}}}. \tag{56}$$

The two $\pm$ symbols in (56) are independent and all four possible choices of signs yield solutions to the quartic. However, only the choice of root with both plus signs will give positive real energies. Since the conversion from (55) to a quartic equation involved squaring, we may have introduced spurious roots and we must check that the purported solution actually satisfies the original equation. Indeed, one finds that substituting the preferred root

$$\widetilde{\mathcal{E}}_{n,m} = \sqrt{\frac{(r-1)(\alpha_n+\beta_m) + \sqrt{4\alpha_n\beta_m - 8r\alpha_n\beta_m + r^2(\alpha_n+\beta_m)^2}}{(1-3r+2r^2)\widetilde{\lambda}}}, \tag{57}$$

into the implicit equation (55) only yields a solution when $r \geq 2$. This behavior is related to the fact that the four roots (56) become degenerate at $r = 2$, coming in two pairs of double roots, and the preferred root only becomes a solution past this crossing point in the region $r \geq 2$.

Another way to interpret this bound is to note that for our high-energy solution (57), one of the two ratios $\frac{4\alpha_n}{r^2|\widetilde{\lambda}|^2\widetilde{\mathcal{E}}_{n,m}^2}$, $\frac{4\beta_m}{r^2|\widetilde{\lambda}|^2\widetilde{\mathcal{E}}_{n,m}^2}$ appearing in (54) tends to $\frac{4}{r^2}$ if $\alpha_n$ is taken to infinity at fixed $\beta_m$, or if $\beta_m \to \infty$ with $\alpha_n$ fixed. In order to guarantee that both ratios remain smaller than 1, so that the arguments of the square roots are positive, we need $r \geq 2$. We conclude that this bound $r \geq 2$ is necessary to have a well-behaved high-energy spectrum in the deformed theory. If one is willing to tolerate a theory with complex energies, there is an interesting phase structure that we see as a function of $r$.

**The $r < 1$ phase**

This phase is morally similar to the bad sign $T\overline{T}$ deformation. Namely, an infinite number of high-energy states have complex energies. More importantly, there are only a finite number of real energies. The easiest way to see this is to look at a plot of energies for a specific $r$. One such case is displayed in Figure 7. Once again, we take $\alpha_0 = \beta_0 = -\frac{1}{4}$ and choose evenly spaced energy levels with a difference of $\frac{1}{10}$. That is, $\alpha_{n+1} - \alpha_n = \frac{1}{10}$ and likewise for the $\beta_m$.

We chose to plot the case $r = \frac{1}{2}$ because this case can also be studied analytically. For the diagonal part of the spectrum ($\alpha_n = \beta_m$), one finds that the implicit equation (30) only admits a real solution if

$$|\alpha| \leq \frac{1}{\widetilde{\lambda}}. \tag{58}$$

This means that the diagonal part of the spectrum has been cut off at high energies. Although the undeformed theory had an infinite tower of states with energies $(\alpha_n, \beta_m)$ with $\alpha_n$ and $\beta_m$ growing arbitrarily large, the corresponding high energy states in the deformed theory either have complex energies or are not present at all.

For states which do satisfy the bound (58), the diagonal deformed energies are given by (50) when $r \neq \frac{1}{2}$. In the special case $r = \frac{1}{2}$, as we mentioned above, the implicit relation degenerates and admits the new solution

$$\mathcal{E} = \frac{2\alpha}{R}, \tag{59}$$

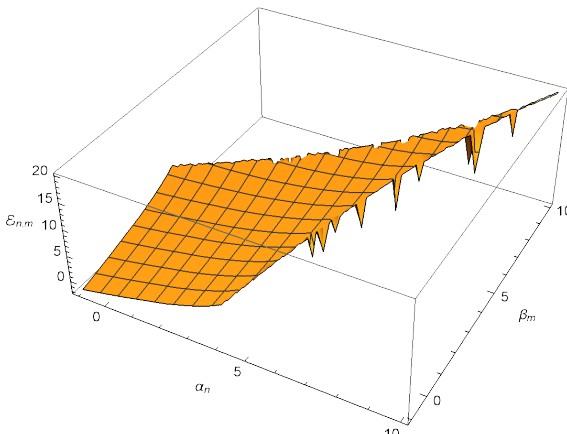

Figure 7: The deformed energies $\mathcal{E}_{n,m}$ as a function of the undeformed dimensionless energies $(\alpha_n, \beta_m)$, and where $r = \frac{1}{2}$, $\widetilde{\lambda} = \frac{1}{10}$. Note that real energies only exist in a finite band around $\alpha_n = \beta_m$ and that solutions fail to exist when $\alpha_n, \beta_m > \frac{1}{\widetilde{\lambda}} = 10$, as expected from equation (58). For undeformed energies outside this region, no real solution exists so the surface plot has been truncated.

which is the same as the corresponding energy level in the undeformed product of CFTs.

In addition to the upper bound in the diagonal sector, we find a second constraint on the difference between the two undeformed energies which must be satisfied in order to give a state in the deformed spectrum. This is most easily seen from a numerical investigation, such as Figure 7 above. We see that, when $|\alpha_n - \beta_m|$ is too large, which corresponds to a point on the plot which is too far from the diagonal, the implicit equation fails to have a real solution. Therefore there is a upper bound on $|\alpha_n - \beta_m|$ in order to have real deformed energies, although the analytic expression for this bound is unwieldy. Graphically, we see that there is a finite ribbon of real energies close to the diagonal which satisfy this bound.

The upshot is that in the range $r < 1$, any interpretation of the deformed spectrum via truncation to a finite number of real energy modes or by some other approach will encounter the same difficulties as bad sign $T\overline{T}$.

**The $1 \le r < 2$ phase**

There is a qualitative change at $r = 1$. In addition to an infinite number of complex energies, there are now an infinite number of real energies, which is unlike the case of bad sign $T\overline{T}$. The real energies are all close to the diagonal case of $\alpha_n = \beta_m$ while the complex energies are far from the diagonal. We can see this numerically in Figure 8. What was a finite ribbon of real energies near the diagonal for $r < 1$ now becomes an ribbon of infinite diagonal extent but finite width.

A simplification occurs in the case of $r = 1$. In this case, the deformed energy levels are

$$\widetilde{\mathcal{E}}_{n,m} = \frac{1}{\widetilde{\lambda}}\left(\sqrt{1 + 2\widetilde{\lambda}(\alpha_n + \beta_m) + \widetilde{\lambda}^2(\alpha_n - \beta_m)^2} - 1\right). \tag{60}$$

However, not every pair of undeformed energies associated with parameters $\alpha_n, \beta_m$ leads to a real energy level in the deformed theory. This is because the implicit equation (30) only admits a real solution for $\widetilde{\mathcal{E}}_{n,m}$ if the parameters satisfy certain bounds. If either $\alpha_n \ge 0$ or $\beta_m \ge 0$, the expression (60) solves (30) with $r = 1$ as long as

$$|\alpha_n - \beta_m| \le \frac{1}{\widetilde{\lambda}}. \tag{61}$$

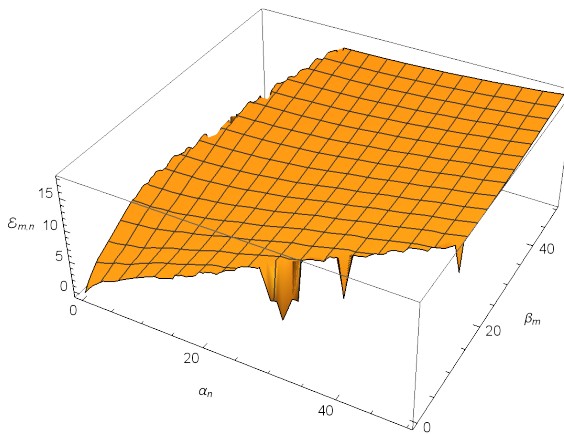

Figure 8: Deformed energies $\mathcal{E}_{n,m}$ for $r = \frac{3}{2}$, $\widetilde{\lambda} = \frac{1}{4}$. In this phase, all of the energies within a finite band around the diagonal $\alpha_n = \beta_m$ remain real in the deformed theory. However, very off-diagonal energies with $|\alpha_n - \beta_m|$ large become complex, or perhaps alternatively become truncated.

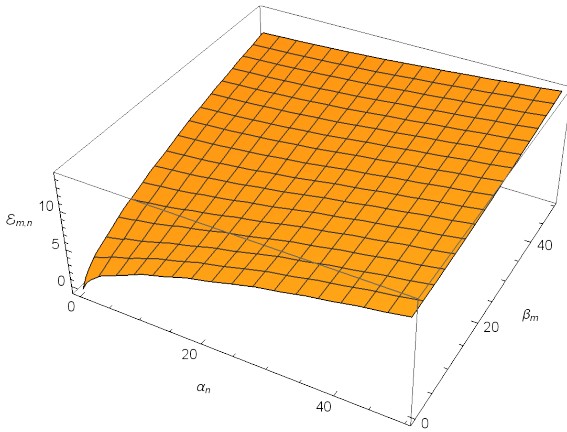

Figure 9: A plot of the deformed energies $\mathcal{E}_{n,m}$ for $r = 2.1$, $\widetilde{\lambda} = \frac{1}{4}$. In the phase $r \geq 2$, all of the deformed energies remain real and there are no complex or truncated energies.

If $\alpha_n$ and $\beta_m$ are both negative, the constraint is

$$|\alpha_n + \beta_m| \leq \frac{1}{2\widetilde{\lambda}}\,. \tag{62}$$

For pairs of undeformed energies which do not satisfy these constraints, there is no real solution to the implicit equation (30).

**The $r \geq 2$ phase**

As mentioned earlier, the case that looks completely real is $r \geq 2$. For comparison with smaller values of $r$, the spectrum is plotted in Figure 9 for $r = 2.1$.

**The ground state**

The other point we want to check is how large $\widetilde{\lambda}$ can become before the ground state energy becomes complex. Many of the analytic results in our preceding discussion assumed diagonal

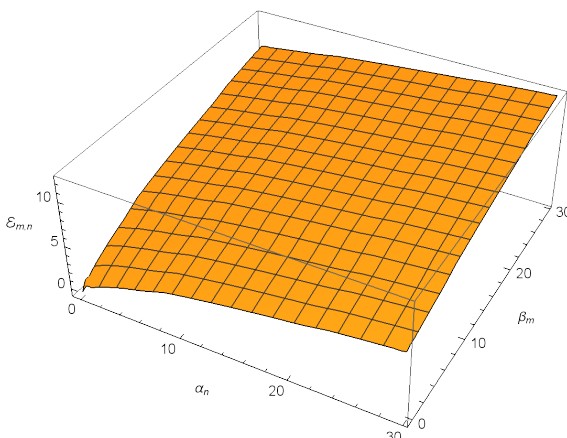

Figure 10: The deformed energies $\widetilde{\mathcal{E}}_{n,m}$ for a combined flow by $\widetilde{\lambda}_1 = \frac{1}{4}$, $\widetilde{\lambda}_2 = 0$, $\widetilde{\lambda}_3 = \frac{1}{4}$. No complex energies arise for this combination of flow parameters.

energies, $\beta_m = \alpha_n = \alpha$, but the spectra of CFT$_1$ and CFT$_2$ were otherwise unconstrained. Now we will assume the ground state energies are the same for both theories so $e_{0,0} = \frac{2\alpha_0}{R}$. From (50), we see that

$$\widetilde{\lambda} \leq \frac{1}{4(2r-1)|\alpha_0|} \, . \tag{63}$$

This is in contrast with (5) where there is an upper bound on $\widetilde{\lambda}$ set by the central charge.

## 3.4 Sequential flows with $\widetilde{\lambda}_2 = 0$

We will briefly mention one additional possibility: one can first deform CFT$_1$ by $\widetilde{\lambda}_1$ then tensor the result with an undeformed CFT$_2$, and finally deform the tensor product by an additional flow with parameter $\widetilde{\lambda}_3$. This corresponds to setting $\widetilde{\lambda}_2 = 0$ in the preceding discussion. Note that bounds for the maximum allowed $\widetilde{\lambda}_3$ when $\widetilde{\lambda}_2 = 0$ were given for some special cases in Table 1. Because the qualitative features of the $\widetilde{\lambda}_2 = 0$ case are similar to the flows described in the preceding subsections, we will not undertake a detailed analysis. Instead we content ourselves with describing the various possibilities and presenting plots to illustrate the behavior.

The first possibility is to deform CFT$_1$ by a positive flow parameter $\widetilde{\lambda}_1 > 0$, tensor with the undeformed CFT$_2$, and then flow the combined system by another positive parameter $\widetilde{\lambda}_3 > 0$. Because this sequential flow involves only positive deformation parameters, we expect it to behave like the all-good-sign flows of section 3.1. These flows appear to produce spectra with all real energies so long as the total length of the positive flows is not so large that the ground state energy becomes complex. An example of the numerical spectrum for a flow of this form is shown in Figure 10, where the undeformed energies in CFT$_1$ and CFT$_2$ are taken to be evenly-spaced with a ground state at $\alpha_0 = \beta_0 = -\frac{1}{4}$ and a gap of 0.1 between adjacent energy levels. Indeed one finds that all of the deformed energies are real.

Another possibility is a sequential flow by $\widetilde{\lambda}_1 < 0$, $\widetilde{\lambda}_2 = 0$, and $\widetilde{\lambda}_3 < 0$. This combination is less interesting because it involves only the bad sign of the deformation parameter and therefore deformed energies corresponding to high-energy states of CFT$_1$ or CFT$_2$ will always be complex.

A more interesting possibility is to first flow by $\widetilde{\lambda}_1 < 0$ and then by $\widetilde{\lambda}_3 > 0$, again with $\widetilde{\lambda}_2 = 0$. We expect this to behave like the sequential flows discussed in section 3.3 where both CFTs were first deformed by the bad sign of the deformation parameter and then the

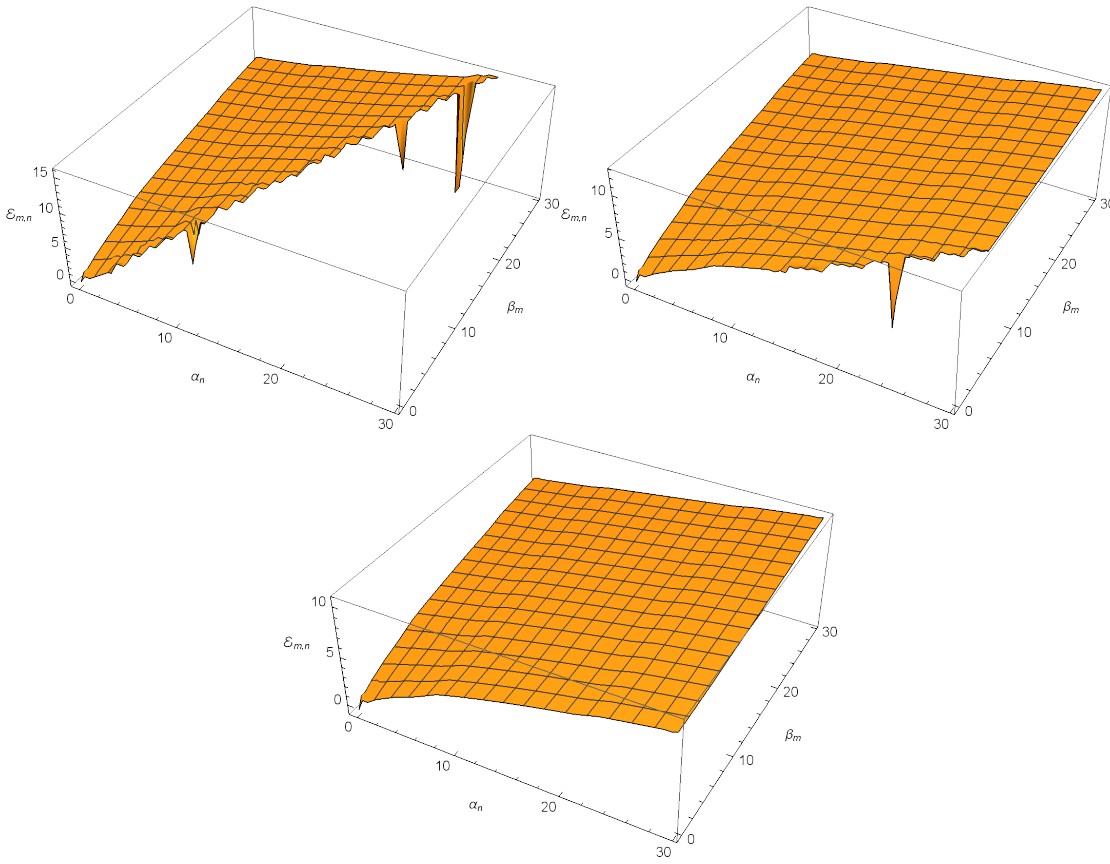

Figure 11: Deformed energies $\widetilde{\mathcal{E}}_{n,m}$ with $\widetilde{\lambda}_1 = -\frac{1}{4}$, $\widetilde{\lambda}_2 = 0$, and $\widetilde{\lambda}_3 = r|\widetilde{\lambda}_1|$. The top-left plot shows $r = 1$ where half of the spectrum below the diagonal becomes complex and is truncated. The top-right plot displays the corresponding spectrum when $r = \frac{3}{2}$; in this case part of the truncated spectrum has been cured, but there is still an infinite region of excised energies below a shifted diagonal which is visible in the bottom-right part of the plot. Finally, the plot on the second line shows the case $r = 2$, where it appears that the entire spectrum has been cured and all deformed energies are real.

combined system was deformed with the good sign. In that context, we saw a phase structure emerge with several possible cases. When the bad-sign flow parameters $\widetilde{\lambda}_1, \widetilde{\lambda}_2$ were too large compared to the good-sign parameter $\widetilde{\lambda}_3$, we saw that part of the spectrum remained complex. When $\widetilde{\lambda}_3$ became sufficiently large the entire deformed spectrum became real. We find numerically that a similar phenomenon occurs when $\widetilde{\lambda}_2 = 0$. In particular, if $\widetilde{\lambda}_1 < 0$ and $\widetilde{\lambda}_3 = r|\widetilde{\lambda}_1|$, we find that part of the deformed spectrum is truncated or becomes complex when $r < 2$, but for $r \geq 2$ all of the deformed energies appear real. This behavior is shown in Figure 11.

Finally, for the class of sequential deformations with $\widetilde{\lambda}_1 > 0$ and $\widetilde{\lambda}_2 = 0$, one could ask about the maximum allowed $\widetilde{\lambda}_3$ with which we may flow before the ground state energy becomes complex. This is the analogue of the question we addressed numerically in Table 1. In the $\widetilde{\lambda}_2 = 0$ case, the approximate maximum values of $\widetilde{\lambda}_3$ and resulting ground state energies for several choices of $\widetilde{\lambda}_1$ are shown in Table 4.

Table 4: For flows with $\widetilde{\lambda}_1 > 0$, $\widetilde{\lambda}_2 = 0$, we have listed the approximate maximum allowed $\widetilde{\lambda}_3$ for which the ground state energy remains real together with the value of that deformed ground state energy.

| $4(\alpha_0, \beta_0)$ | $\widetilde{\lambda}_1$ | Max $\widetilde{\lambda}_3$ | 4×Energy |
|---|---|---|---|
| $(-1, -1)$ | 0.25 | 0.41 | $-3.42$ |
| $(-1, -1)$ | 0.5 | 0.302 | $-3.41$ |
| $(-1, -1)$ | 0.9 | 0.006 | $-2.9$ |
| $(-1, -2)$ | 0.25 | .29 | $-5.3$ |
| $(-1, -2)$ | 0.5 | 0.21 | $-4.54$ |
| $(-1, -2)$ | 0.9 | 0.0049 | $-3.55$ |

## 4 Flow Equation for the Lagrangian

In the preceding sections, we have considered the flow equation for the energy levels in a pair of CFTs coupled via a $T_1 T_2$ procedure. Aside from the checks of modular invariance, much of the discussion is restricted to the zero-momentum sector where we can use an implicit solution to the inviscid Burgers' equation. Next we will consider the flow equation for the Lagrangian itself. This is a classical analysis so we are not constrained by quantum constraints on whether the theory is sensible. For the original $T\overline{T}$ deformation, the study of classical flows has led to an unexpected new organizing principle for many well-known effective actions [3, 28–36]. Along the first leg of our multi-step deformation, the Lagrangian obeys the differential equation:

$$\frac{\partial \mathcal{L}_\lambda}{\partial \lambda} = \frac{1}{2}\left(\left(T^\mu_{\ \mu}\right)^2 - T^{\mu\nu} T_{\mu\nu}\right). \tag{64}$$

### 4.1 Two Free Bosons

We will be primarily motivated by the example of two free bosons $\phi$ and $\chi$, with an initial seed theory of the form

$$\mathcal{L}_0 = \partial^\mu \phi \partial_\mu \phi + \partial^\mu \chi \partial_\mu \chi. \tag{65}$$

For the first step of our deformation, we will separately deform $\text{CFT}_1$ of the scalar $\phi$ by $T_1 \overline{T}_1$ and deform $\text{CFT}_2$ of the scalar $\chi$ by $T_2 \overline{T}_2$, both by a total parameter $\lambda$. It was first shown in [3] that this procedure of deforming a free scalar produces a deformed Lagrangian corresponding to a Nambu-Goto string in static gauge. After the first leg of our deformation, the resulting theory is therefore simply the tensor product of two gauge-fixed Nambu-Goto theories:

$$\mathcal{L}_\lambda = \frac{1}{2\lambda}\left(\sqrt{1 + 2\lambda \partial^\mu \phi \partial_\mu \phi} + \sqrt{1 + 2\lambda \partial^\mu \chi \partial_\mu \chi} - 2\right). \tag{66}$$

We would now like to consider (66) as a new seed theory and deform by the total $T\overline{T}$ of the combined theory. In a sense, the first flow by $\lambda$ takes us from point particles to gauge-fixed strings. The second flow should be taking us to a kind of interacting theory of multiple strings.

As a first step in the analysis, it will be useful to consider the possible scalar quantities that can appear in the final Lagrangian after performing this deformation. The Hilbert stress tensor $T_{\mu\nu}$ of (66) contains one term proportional to $\partial_\mu \phi \partial_\nu \phi$, one term proportional to $\partial_\mu \chi \partial_\nu \chi$, and

one term of the form $g_{\mu\nu}\mathcal{L}_\lambda$. When we construct bilinears in $T_{\mu\nu}$, therefore, three independent Lorentz scalars will appear:

$$x = \partial^\mu\phi\,\partial_\mu\phi\,, \qquad y = \partial^\mu\chi\,\partial_\mu\chi\,, \qquad z = \partial^\mu\phi\,\partial_\mu\chi\,. \tag{67}$$

It is clear that this exhausts the list of scalars that can be constructed from $\partial^\mu\phi$ and $\partial^\mu\chi$, since we will never generate terms with more than one derivative per field and every index appearing on a derivative $\partial^\mu$ of a field must appear contracted with a derivative $\partial_\mu$ of another field. During the second step of the flow, we can therefore assume that the Lagrangian takes the form

$$\mathcal{L}_{\lambda,\lambda_3} = f(\lambda, \lambda_3, x, y, z)\,, \tag{68}$$

where we use the symbol $\lambda_3$ for the deformation parameter along the second leg of the deformation. We write $\mathcal{L}_{\lambda,\lambda_3}$ for the final deformed Lagrangian. The Hilbert stress tensor associated with $\mathcal{L}_{\lambda,\lambda_3}$ is

$$T_{\mu\nu} = -2\left(f_x\,\partial_\mu\phi\,\partial_\nu\phi + f_y\,\partial_\mu\chi\,\partial_\nu\chi + f_z\,\partial_{(\mu}\phi\,\partial_{\nu)}\chi\right) + g_{\mu\nu}f\,, \tag{69}$$

where we write $f_x = \frac{\partial f}{\partial x}$ and so on. Using (69) we can construct the bilinears appearing in (64). First the trace is

$$T^\mu{}_\mu = -2\left(x\,f_x + y\,f_y + z\,f_z\right) + 2f\,, \tag{70}$$

and the contraction $T_{\mu\nu}T^{\mu\nu}$ is

$$\begin{aligned}
T_{\mu\nu}T^{\mu\nu} = {}&4\left(x^2f_x^2 + 2z^2f_xf_y + 2xzf_xf_z + y^2f_y^2 + 2yzf_yf_z + \frac{1}{2}(xy+z^2)f_z^2\right)\\
&-4f\left(xf_x + yf_y + zf_z\right) + 2f^2\,.
\end{aligned} \tag{71}$$

The flow equation with respect to the $\lambda_3$ variable is therefore

$$\begin{aligned}
\frac{\partial\mathcal{L}_{\lambda,\lambda_3}}{\partial\lambda_3} &= \frac{1}{2}\left(\left(T^\mu{}_\mu\right)^2 - T^{\mu\nu}T_{\mu\nu}\right)\\
&= f^2 - 2f\left(x\,f_x + y\,f_y + z\,f_z\right) + (4f_xf_y - f_z^2)(xy - z^2)\,.
\end{aligned} \tag{72}$$

Starting from the seed theory (66), we can use this flow equation to find a perturbative solution to any desired order in $\lambda_3$. For instance, up to $\mathcal{O}(\lambda_3)$, one has

$$\begin{aligned}
\mathcal{L}_{\lambda,\lambda_3} = {}&\mathcal{L}_{\lambda,0} + \frac{3\lambda_3}{2\lambda^2} + \frac{\lambda_3}{2\lambda^2\sqrt{(1+2\lambda x)(1+2\lambda y)}}\Bigg[1 + \lambda(x+y) + 2\lambda^2(xy-z^2)\\
&-2\left(\sqrt{1+2\lambda x} + \sqrt{1+2\lambda y}\right) - 2\lambda\left(y\sqrt{1+2\lambda x} + x\sqrt{1+2\lambda y}\right)\Bigg] + \mathcal{O}(\lambda_3^2)\,,
\end{aligned} \tag{73}$$

where for convenience we repeat

$$\mathcal{L}_{\lambda,0} \equiv \mathcal{L}_\lambda = \frac{1}{2\lambda}\left(\sqrt{1+2\lambda\partial^\mu\phi\,\partial_\mu\phi} + \sqrt{1+2\lambda\partial^\mu\chi\,\partial_\mu\chi} - 2\right)\,. \tag{74}$$

Expanding (73) to leading order in $\lambda$ gives

$$\begin{aligned}
\mathcal{L}_{\lambda,\lambda_3} = {}&\mathcal{L}_{\lambda,0} + \lambda_3\left(xy - z^2 - \frac{1}{4}(x+y)^2\right) + \lambda_3\lambda(x+y)\left(\frac{1}{2}(x^2+y^2) - (xy-z^2)\right)\\
&+ \mathcal{O}(\lambda_3^2, \lambda^2\lambda_3)\,.
\end{aligned} \tag{75}$$

The $\mathcal{O}(\lambda_3\lambda^0)$ term of (75) reproduces the leading contribution when $\lambda = 0$. This corresponds to deforming the tensor product of two free bosons by a $T\overline{T}$ deformation of the total system. The exact closed form for this case was presented in [3] and corresponds to a gauge-fixed Nambu-Goto string with two transverse directions rather than one. The deformed action with $\lambda = 0$ has an $O(2)$ symmetry rotating $\phi$ into $\chi$. The combinations $xy - z^2$ and $(x + y)^2$ separately respect this symmetry.

The $\mathcal{O}(\lambda_3\lambda)$ term, however, explicitly breaks this symmetry as we expect since the action (66) does not respect this symmetry. Finding a closed form solution for the flow equation appears to be difficult in this model by contrast with cases like [37, 38] where exact implicit solutions were possible. Knowing the exact form would be very interesting for cases like $\lambda < 0$ followed by $\lambda_3 > 0$ where we expect the bad behavior of a string with negative tension to be cured by the forward flow. It is natural to suspect that there is a critical velocity for such models similar to the critical velocity seen in (66) with the good sign flow $\lambda > 0$.

## 4.2 Two Free Fermions

An even simpler case to consider is the $T_1 T_2$ coupling of two free Majorana fermions, since the number of allowed terms is severely constrained by nilpotency. Consider two fermionic fields $\psi_\pm$ and $\zeta_\pm$ with the undeformed action

$$\mathcal{L}_0 = i\psi_+ \partial_{--}\psi_+ + i\psi_- \partial_{++}\psi_- + i\zeta_+ \partial_{--}\zeta_+ + i\zeta_- \partial_{++}\zeta_- . \tag{76}$$

Here we use bispinor notation for vector indices; for details on these conventions, see [30,31], for example. Next we would like to compute the components of the stress tensor. Using the usual Noether procedure but being careful to account for Grassmann statistics, one finds that the stress tensor components of a general fermionic theory for a single field $\psi_\pm$ are given by

$$\begin{aligned}
T_{++++} &= (\partial_{++}\psi_+)\frac{\delta\mathcal{L}}{\delta(\partial_{--}\psi_+)} + (\partial_{++}\psi_-)\frac{\delta\mathcal{L}}{\delta(\partial_{--}\psi_-)}\,, \\
T_{++--} &= (\partial_{--}\psi_+)\frac{\delta\mathcal{L}}{\delta(\partial_{--}\psi_+)} + (\partial_{--}\psi_-)\frac{\delta\mathcal{L}}{\delta(\partial_{--}\psi_-)} - \mathcal{L}\,, \\
T_{--++} &= (\partial_{++}\psi_+)\frac{\delta\mathcal{L}}{\delta(\partial_{++}\psi_+)} + (\partial_{++}\psi_-)\frac{\delta\mathcal{L}}{\delta(\partial_{++}\psi_-)} - \mathcal{L}\,, \\
T_{----} &= (\partial_{--}\psi_+)\frac{\delta\mathcal{L}}{\delta(\partial_{++}\psi_+)} + (\partial_{--}\psi_-)\frac{\delta\mathcal{L}}{\delta(\partial_{++}\psi_-)}\,.
\end{aligned} \tag{77}$$

Note that the Noether stress tensor is not symmetric ($T_{++--} \neq T_{--++}$) which is a generic feature of theories with fermions. It can be made symmetric via an improvement transformation or by using the appropriate version of the Hilbert stress tensor, but for our purposes the Noether stress tensor will be sufficient. The analogous stress tensor components for the subsystem with the field $\zeta_\pm$ can be obtained by replacing $\psi_\pm$ with $\zeta_\pm$ in (77).

Using these expressions for the components of $T_{\mu\nu}$, it is straightforward to find the $T\overline{T}$-deformed Lagrangian after deforming each fermion theory separately by $\lambda$,

$$\mathcal{L}_\lambda = \mathcal{L}_0 + \lambda\Big(\psi_+ \partial_{--}\psi_+ \psi_- \partial_{++}\psi_- - \psi_+ \partial_{++}\psi_+ \psi_- \partial_{--}\psi_- + \zeta_+ \partial_{--}\zeta_+ \zeta_- \partial_{++}\zeta_- - \zeta_+ \partial_{++}\zeta_+ \zeta_- \partial_{--}\zeta_-\Big). \tag{78}$$

Although (78) only contains a correction which is linear in $\lambda$, it actually satisfies the exact $T\overline{T}$ flow (within each sector) to all orders in $\lambda$, because all higher terms vanish. This is simply because there are no additional non-vanishing terms that can be constructed from a single fermion because of Grassmann statistics.

Now we would like to treat (78) as a seed theory and $T\overline{T}$ deform again but this time with parameter $\lambda_3$. The components of the stress tensor associated with (78), using bispinor

conventions for the vector indices, are

$$
\begin{aligned}
T_{++++} &= i\psi_+\partial_{++}\psi_+ + i\zeta_+\partial_{++}\zeta_+, \\
T_{++--} &= -i\psi_-\partial_{++}\psi_- - i\zeta_-\partial_{++}\zeta_-, \\
T_{--++} &= -i\psi_+\partial_{--}\psi_+ - i\zeta_+\partial_{--}\zeta_+, \\
T_{----} &= i\psi_-\partial_{--}\psi_- + i\zeta_-\partial_{--}\zeta_-.
\end{aligned} \tag{79}
$$

In particular, the $\mathcal{O}(\lambda)$ term drops out of the stress tensor entirely. Therefore the two-step deformed Lagrangian is

$$
\begin{aligned}
\mathcal{L}_{\lambda,\lambda_3} =\,& \mathcal{L}_0 + (\lambda + \lambda_3)\big(\psi_+\partial_{--}\psi_+\psi_-\partial_{++}\psi_- - \psi_+\partial_{++}\psi_+\psi_-\partial_{--}\psi_- + \zeta_+\partial_{--}\zeta_+\zeta_-\partial_{++}\zeta_- \\
& - \zeta_+\partial_{++}\zeta_+\zeta_-\partial_{--}\zeta_-\big) + \lambda_3\big(\psi_+\partial_{--}\psi_+\zeta_-\partial_{++}\zeta_- - \psi_+\partial_{++}\psi_+\zeta_-\partial_{--}\zeta_- \\
& + \zeta_+\partial_{--}\zeta_+\psi_-\partial_{++}\psi_- - \zeta_+\partial_{++}\zeta_+\psi_-\partial_{--}\psi_-\big).
\end{aligned} \tag{80}
$$

One can verify by direct computation that the components of the stress tensor associated with $\mathcal{L}_\lambda$ are identical to those associated with $\mathcal{L}_{\lambda,\lambda_3}$. Thus (80) gives the exact, all-orders solution in both $\lambda$ and $\lambda_3$ to the two-step flow. For instance, we can set $\lambda = -\lambda_3$ to find

$$
\begin{aligned}
\mathcal{L}_{-\lambda,\lambda} =\,& \mathcal{L}_0 + \lambda\big(\psi_+\partial_{--}\psi_+\zeta_-\partial_{++}\zeta_- - \psi_+\partial_{++}\psi_+\zeta_-\partial_{--}\zeta_- \\
& + \zeta_+\partial_{--}\zeta_+\psi_-\partial_{++}\psi_- - \zeta_+\partial_{++}\zeta_+\psi_-\partial_{--}\psi_-\big),
\end{aligned} \tag{81}
$$

which gives the finite-$\lambda$ solution for the $T_1\overline{T}_2$ deformation discussed in (7). As mentioned above, this is an irrelevant coupling of the two seed theories but the deforming operator

$$
\begin{aligned}
\mathcal{O}_{T_1\overline{T}_2} &\equiv \frac{\partial \mathcal{L}_{-\lambda,\lambda}}{\partial \lambda} \\
&= \psi_+\partial_{--}\psi_+\zeta_-\partial_{++}\zeta_- - \psi_+\partial_{++}\psi_+\zeta_-\partial_{--}\zeta_- + \zeta_+\partial_{--}\zeta_+\psi_-\partial_{++}\psi_- - \zeta_+\partial_{++}\zeta_+\psi_-\partial_{--}\psi_-,
\end{aligned} \tag{82}
$$

cannot be expressed as any scalar quantity constructed from the stress tensor $T_{\mu\nu}$ of the theory. It is not the product of currents and is therefore a qualitatively different deformation from any $T\overline{T}$-like deformation.

Although this solution for the Lagrangian of the $T_1\overline{T}_2$ deformed system has a fairly simple form involving four fermion interactions, it is interesting to note that the cylinder spectrum of this interacting theory is still in principle determined by iterated applications of the inviscid Burgers' equation, as we discussed in section 3.

## Acknowledgements

We would like to thank Anthony Ashmore, Nima Afkhami-Jeddi and David Kutasov for helpful discussions. S. S. is supported in part by NSF Grant No. PHY1720480 and PHY2014195. C. F. is supported by U.S. Department of Energy grant DE-SC0009999 and by funds from the University of California.

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
