# Peer review of "Sequential Flows by Irrelevant Operators"

_SciPost Physics, doi:SciPost Phys. 14, 098 (2023)_

## Round 1 · Referee Report · Anonymous (Referee 1) · 2022-10-11

Strengths

1- The authors propose a creative and interesting generalization of the known $T \overline T$ deformations. 2- They perform an extensive analysis, both analytical and numerical, of many of the qualitatively different cases their deformation allows. 3- This analysis sheds some light on the complex energy levels, one of the big open questions related to the $T \overline T$ deformation.

Weaknesses

1- It is not clear whether some of the bounds are optimal, see full report. 2- Despite mentioning the holographic interpretation in the abstract, the authors are unfortunately rather brief and noncommittal in their conclusion on this aspect.

Report

The authors study an analog of the $T \overline T$ deformation obtained by deforming a pair of, already deformed, field theories by the operator $(T_1 + T_2) (\overline T_1 + \overline T_2)$. They analyze extensively the energy spectrum of these theories for several values of each of the individual deformations. Notably, it turns out that one can "recover" states whose energy had become complex after the first $T \overline T$ deformation by applying a judicially chosen "joint" $T \overline T$ deformation. The authors furthermore give numerical evidence that these theories remain invariant under S-duality.
They conclude by studying the classical Lagrangian for some examples and comment briefly on possible holographic interpretations.

This paper presents an interesting perspective on an important unanswered question for the $T \overline T$ deformation, namely the faith of the complex energy states. The analysis suggests that these energies become real again by a reverse $T \overline T$ deformation, or even, more surprisingly, by different deformations.

That said, I would like to add some remarks and questions. First off, I think that section 2 would benefit from a precise clarification of what the nontrivial statement is. It does not seem surprising that, within a certain range of parameters, a $T \overline T$ deformation can be undone by a reverse deformation because the spectrum is given by the Burger's flow, which can be solved by the method of characteristics and is reversible as long as a unique solution exists, i.e. as long as the characteristics do not intersect. It seems therefore that the nontrivial claim in section 2 is that the flow is reversible even beyond this point, if one specifies a finite amount of data such as the specific branch of the square root that is chosen. Is this a fair summary, and could the authors provide more comments?

Second, it is surprising that the bound $r < 1/2$ from the diagonal spectrum (3.22) is much stronger than the bound $r < 1$ from the off-diagonal spectrum (3.24). This seems to suggest that (3.8), which was derived by first making a large energy approximation and then flowing with the joint $T \overline T$ deformation, is not an accurate estimate. One can for example compare (3.8) with the exact result when $\alpha = \beta$, $\lambda_1 = \lambda_2 = \lambda$ and $\lambda_3 = - r \lambda$ for $r \approx 1/2$, these do not seem to match.

Finally, it would be very interesting if the authors could provide some more precise comments on the holographic correspondence they have in mind.
  • validity: good
  • significance: good
  • originality: high
  • clarity: good
  • formatting: excellent
  • grammar: perfect

Author:  Christian Ferko  on 2022-11-04  [id 2982]

(in reply to Report 1 on 2022-10-11)

We thank the referee for his or her comments.

First, in response to the suggestions about Section 2, we have clarified the wording to emphasize that this discussion is a warm-up example which is meant to illustrate some of the phenomena of later sections in a well-understood context. There is no claim of any novel result for this case of a single system.

Next, to clarify the referee's query about the bounds: the strong bound in equation (3.22) is for the case of good sign followed by bad sign, while equation (3.8) is for the case of good sign followed by good sign. This is the reason why those two bounds are so different.

Finally, we completely agree that applications to holography need to be explored more fully. However, that exploration probably merits a separate discussion since this work is largely focused on defining sequential deformations from a field theory perspective.

---

## Round 1 · Referee Report · Anonymous (Referee 2) · 2022-10-16

Strengths

1 - The paper is well written
2 - The studied setup is original

Weaknesses

1 - The holographic interpretation is fairly speculative
2 - In various points, the authors have to resort to numerical analysis

Report

This manuscript deals with theories obtained by applying sequential $T\bar{T}$ deformations.

In Section 2, the authors consider the case where a single CFT is subject to two consecutive $T\bar{T}$ deformations with parameters $λ_1$ and $λ_2$. They observe that in the zero-momentum sector, the spectrum is the same as if obtained through a single deformation with parameter $λ_1+λ_2$.

In Section 3, the authors study the deformation of a theory obtained by tensoring two CFTs that were previously separately deformed. The resulting theory is made non-trivial thanks to the non-linear nature of the $T\bar{T}$ operator, introducing an interaction between the two sectors.
The analysis is, in various points, based on numerical evidence.

In Section 4, the same setup of Section 3 is considered from a Lagrangian standpoint. Specifically, the authors consider as a starting point a theory of two free scalars or, alternatively, two free fermions.

By the authors' own admission, this work is somewhat explorative in nature, but the results presented seem to me correct. However, one of the main points studied in the manuscript concerns what happens when one combines deformations with opposite signs. A $T\bar{T}$ deformation with negative $λ$ acting on a CFT is known to induce a complexification of the energy eigenvalues for some upper portion of the spectrum, leaving only a finite number of real energy levels.
To understand to which extent a second deformation with positive $λ$ can, at least partially, restore the real spectrum one should first understand how to properly interpret a deformation with a negative parameter in the first place. Do the complex energies signal a non-unitarity of the theory? Should they be removed from the spectrum? Is the deformed finite-volume theory even well-defined?
Given that the effect of a single deformation is still not fully understood from a physical standpoint, it is difficult to draw any conclusion regarding the outcome of sequential deformations.

Furthermore, the results of Section 2 seem to me just an obvious consequence of the very definition of the $T\bar{T}$ deformation, where at any point $λ$ along the flow, the infinitesimal deformation is induced by the $T\bar{T}$ operator computed at $λ$. Finite deformations $g_\lambda$, thus, act as elements of a one-parameter group. Schematically: $g_{λ_1} \cdot g_{λ_2} = g_{λ_1+λ_2}$.
In fact, this property should hold for general two-dimensional theories (i.e. not just conformal field theories), and it is not at all restricted to the zero-momentum sector.
While the authors seem to briefly hint at this fact in the Introduction, they then proceed to derive all the results by studying certain solutions of the flow equation. Yet, it seems to me that, being familiar with how a single $T\bar{T}$ deformation acts on CFTs, one could anticipate the results of Section 2 without performing any explicit computation.

I recommend publication, provided the authors address the points raised below:

Requested changes

1 - It is not clear to me what Eq. (1.6) represents. It is supposed to be the first order term in $λ_i$ of the deforming operator, which in principle should be obtained by expanding $$\lambda_3 (T_1(\lambda_1)+T_2(\lambda_2)) (\bar{T}_1(\lambda_1)+\bar{T}_2(\lambda_2))$$ at the first order in $λ_1$ and $λ_2$. In Eq. (1.7), the case $λ_3=-λ_1=-λ_2$ is considered. However, the flows induced by $λ_1$ and $λ_2$ do not commute with the flow induced by $λ_3$. Therefore, as indicated in Figure 1, when $λ_3$ is small, $λ_1$ and $λ_2$ should be finite.
The authors should clarify this paragraph.
Similar considerations apply to Eq. (3.23).

2 - The authors should stress that the results of Section 2 are, as one would expect, consistent with a single deformation and that this is a general fact.

3 - There is a typo in Eq. (1.6), where parentheses do not match.

4 - In Eq. (3.5), $\mathcal{E}_n$ should read $\mathcal{E}_{m,n}$.

  • validity: good
  • significance: ok
  • originality: good
  • clarity: high
  • formatting: excellent
  • grammar: perfect

Author:  Christian Ferko  on 2022-11-04  [id 2983]

(in reply to Report 2 on 2022-10-16)

We thank the referee for his or her comments.

First, in response to the suggestions about Section 2, we have clarified the wording to emphasize that this discussion is a warm-up example which is meant to illustrate some of the phenomena of later sections in a well-understood context. There is no claim of any novel result for this case of a single system.

The referee brings up an interesting comment about whether theories with complex energies are in any sense sensible. This is a point over which we ruminated a great deal. In this work, we have taken the attitude that there might exist a prescription for treating such theories, but the cases we call sensible actually have no complex energies at all. We use that criterion to define our good theories.

It can be misleading to think about the sequential flows in terms of the spectrum seen at intermediate steps. Much like the single-system case reviewed in Section 2, where a good sign flow can be viewed as a combination of bad and good sign flows, the final spectrum is completely sensible.

We have fixed the typo in equation (1.6). More importantly, we have clarified the definition of the leading irrelevant operator. We agree that the previous $\lambda$-dependence could introduce confusion about the non-commutativity of the flows.

We thank the referee for pointing out the typo in equation (3.5), which is fixed.

Finally, we completely agree that applications to holography need to be explored more fully. However, that exploration probably merits a separate discussion since this work is largely focused on defining sequential deformations from a field theory perspective.

---

## Round 2 · Referee Report · Anonymous (Referee 3) · 2022-11-21

Report

I would like to thank the authors for their kind reply and clarifications in the paper.

Unfortunately, I believe that my remark about equations (3.22), (3.24) and (3.8) was misunderstood. The point was not to compare (3.8) to (3.22), but rather to explain why the authors obtained a different bound on $r$ from (3.24) than they did from (3.22). The reason is that to obtain (3.24), they used the same methodology as in (3.8), but I believe equation (3.8) is wrong.

To see this explicitly, one can check (3.8) in a special case, namely $\lambda_2 = \lambda_1$ and $\beta = \alpha$. The approximation (3.8) than reduces to $$\mathcal{E}_{n, m} \sim \frac{2 \sqrt{\alpha \lambda_1} - R}{\lambda_1 + \lambda_3} \ .$$ However, we can obtain the exact result for $\lambda_2 = \lambda_1$ and $\beta = \alpha$ by starting from (3.6): $$\mathcal{E}_{n, m} = \frac{R + \lambda_3 \mathcal{E}_{n, m}}{\lambda_1} \left( \sqrt{1 + \frac{4\lambda_1 \alpha_n}{(R + \lambda_3 \mathcal{E}_{n,m})^2}} - 1 \right) \ ,$$ which can be solved to give $$\mathcal{E}_{n, m} = \frac{-R + \sqrt{R^2 + 4 \alpha \lambda_1 + 8 \alpha \lambda_3}}{\lambda_1 + 2 \lambda_3} \ .$$ This result is exact and does not agree with the approximation from (3.8). In particular, the numerators between the prediction from (3.8) and the exact result differ with a factor of 2 in front of $\lambda_3$. I believe this is the same factor of 2 that distinguishes the bound $r < 1$ obtained in (3.24) from the actual bound $r < 1/2$ from (3.22).

What went wrong is that an implicit equation for $\mathcal{E}_{n,m}$ was used after the large $\alpha$ approximation (3.7), but $\mathcal{E}_{n,m}$ itself is of order $\sqrt{\alpha}$. Presumably one can obtain a correct version of (3.8) by taking into account the next order in (3.7).

Unless the authors disagree with this argument, I think it best to address this issue before proceeding with the publication of the paper.

Requested changes

Address the issue with (3.8) and (3.24).

  • validity: -
  • significance: -
  • originality: -
  • clarity: -
  • formatting: -
  • grammar: -

Author:  Christian Ferko  on 2022-12-09  [id 3117]

(in reply to Report 1 on 2022-11-21)

We thank the referee for his or her comments. The issue the referee raised is now clear to us. The resolution is interesting, and we have revised the discussion around (3.7) - (3.8). It turns out that the formulae we originally listed were correct but only in a perturbative expansion in $\lambda_3$ around $0$. The discrepancy with the later formula goes away when one performs this expansion to leading order in $\lambda_3$. We have also revised any subsequent discussion that relied on those results.

---

## Round 2 · Referee Report · Anonymous (Referee 4) · 2022-12-5

Report

I would like to thank the authors for their kind reply.
I believe they addressed the points I raised in a satisfactory way, with possibly the exception of point 1.

I still find the concept of a leading deformation in the context of this paper quite confusing.
The considered setup is obtained by applying two distinct and consecutive finite deformations and cannot be understood as being generated by a single irrelevant operator.
In fact, the deformed theory is controlled by three parameters, namely $\{λ_1,λ_2, λ_3\}$, and there is no point along the flow where these are all small. One can insist on studying such a regime, but it is probably just misleading. For instance, I don't see how the operator in Eq. (1.7) should be in any way related to the theory with finite $λ_3 = -λ_1 = -λ_2$, yet the authors state that "[...] the procedure of sequentially deforming that we described would seem to define some theory, whose leading order deformation is (1.7)."

Requested changes

1 - The authors should clarify the paragraph mentioned above.

  • validity: -
  • significance: -
  • originality: -
  • clarity: -
  • formatting: -
  • grammar: -

Author:  Christian Ferko  on 2022-12-09  [id 3116]

(in reply to Report 2 on 2022-12-05)

We would like to thank the referee for his or her comments. We have gone ahead and revised the paragraph for improved clarity concerning this point.

---

## Round 3 · Referee Report · Anonymous (Referee 2) · 2022-12-22

Report

The authors addressed all the points I raised in my previous reports. I recommend publication.

---

## Round 3 · Referee Report · Anonymous (Referee 1) · 2022-12-29

Report

I thank the authors for adding the subleading term in (3.7) and (3.8). The additional terms did not end up changing the bound (3.24), which suggests that it might be substantially more difficult to obtain a sharper bound than the one derived here. That is outside the reasonable scope for this article.

Given the strengths of this paper already mentioned in the review of version 1, I recommend publication as is.

---

## Round 3 · Author Response

We are resubmitting the manuscript after making the second round of revisions suggested by the two referees.

---

## Round 3 · List of Changes

The discussion of the leading irrelevant operator around equation (1.7) has been clarified. The derivation arounds equations (3.7) and (3.8) has been corrected and the language now emphasizes that one must work to leading order in $\lambda_3$ in intermediate steps for the last line of equation (3.8) to be correct. The sentences before and after equation (3.24) has also been modified, since this result depended on equation (3.8), to reflect the additional assumption that $\lambda_3$ and thus $r$ are small.

---

## Editorial Decision

published